# α-Amidoamids as New Replacements of Antibiotics—Research on the Chosen K12, R2–R4 *E. coli* Strains

**DOI:** 10.3390/ma13225169

**Published:** 2020-11-16

**Authors:** Paweł Kowalczyk, Arleta Madej, Mateusz Szymczak, Ryszard Ostaszewski

**Affiliations:** 1Department of Animal Nutrition, The Kielanowski Institute of Animal Physiology and Nutrition, Polish Academy of Sciences, Instytucka 3, 05-110 Jabłonna, Poland; 2Institute of Organic Chemistry, Polish Academy of Sciences, Kasprzaka 44/52, 01-224 Warsaw, Poland; arleta.madej@icho.edu.pl; 3Department of Molecular Virology, Institute of Microbiology, Faculty of Biology, University of Warsaw, Miecznikowa 1, 02-096 Warsaw, Poland; mszymczak@biol.uw.edu.pl

**Keywords:** oxidative stress, Ugi reaction, peptidomimetics, Fpg-*N*-glycosylase/AP lyase DNA, LPS—lipopolysaccharide, alpha-amidoamides (AAAs)

## Abstract

A preliminary study of α-amidoamids as new potential antimicrobial drugs was performed. Special emphasis was placed on selection of structure of α-amidoamids with the highest biological activity against different types of Gram-stained bacteria by lipopolysaccharide (LPS). Herein, *Escherichia coli* model strains K12 (without LPS in its structure) and R1–R4 (with different length LPS in its structure) were used. The presented work showed that the antibacterial activity of α-amidoamids depends on their structure and affects the LPS of bacteria. Moreover, the influence of various newly synthesized α-amidoamids on bacteria possessing smooth and rought LPS and oxidative damage of plasmid DNA caused by all newly obtained compounds was indicated. The presented studies clearly explain that α-amidoamids can be used as substitutes for antibiotics. The chemical and biological activity of the analysed α-amidoamids was associated with short alkyl chain and different isocyanides molecules in their structure such as: *tetr*-butyl isocyanide or 2,5-dimethoxybenzyl isocyanide. The observed results are especially important in the case of the increasing resistance of bacteria to various drugs and antibiotics.

## 1. Introduction

Peptides are essential for almost every physiological process in the cell. Small peptides induced by endogenous factors such as hormones or exogenous e.g., active substances such as toxins have various biological activities [1,2,3,4].

Such molecules, called peptidomimetics, appear to be an excellent starting material for the development of new candidate drugs that can mimic the structure or activity of natural peptides. They contain appropriately chemically modified analogues synthesized on the basis of neuropeptides and peptide hormones, showing species-specific and organ-selective, directed physiological action, which significantly increases their stability under metabolic conditions, but retains biological activity [1,2,3,4]. The molecular goal of peptidomimetics is the interaction with receptor ligands, inhibitors of protein-protein interaction, and enzyme inhibitors. These compounds, compared to normal peptides, show greater stability against proteolysis and bioavailability of the receptor in biological processes [1,2,3,4]. Furthermore, the hydrophilic character of peptides limits their permeability through biological membranes. Methods for the synthesis of natural peptides are well known but the synthesis of the peptidomimetics requires more complex methodology [5,6,7]. Chemical modifications in the analysis of small molecule peptidomimetics include limiting their conformation by cyclization of peptides or by including unnatural amino acids and dipeptide substitutes [8,9,10,11,12]. This is done by replacing a specific peptide bond with an isoster. A lot of synthetic methods that are based on multistep procedures are not appropriate for searching new drug candidates. For this purpose, the use of multicomponent of Ugi reaction is required, which gives the opportunity to synthesize a huge amount of new chemical compounds, using various ketones, aldehydes, amine, isonitriles, and carboxylic acid to form an amide [13,14,15,16,17,18,19,20,21,22,23,24,25,26]. The reaction is exothermic and its rate depends on the rate of isonitrile addition. The obtained substances may prove useful in the development of new pharmaceutically active substances [13,14,15,16,17,18,19,20,21,22,23,24,25,26] and may be synthesized by the Ugi reaction (Figure 1). Examples include protease inhibitors (Telaprevir, Crixivan, Indavir—against HIV) or kainic anesthetics (lidocaine, bupivacaine) [27,28,29,30,31,32,33,34,35,36,37,38,39,40,41,42,43,44]. They can be used successfully in the treatment of bacterial infections, viral diseases, cardiovascular diseases, diabetes, osteoporosis, and neoplasms, with the use of calpain inhibitors and the proteasome-inhibiting drug Bortezomib (Figure 1) [27,28,29,30,31,32,33,34,35,36,37,38,39,40,41,42,43,44].

However, there are some disadvantages to using peptides and protein fragments as therapeutic agents related to the rapid degradation of peptides by proteases or lipases present in the cell [1,2,3,4]. According to the literature, slight chemical modifications of Ugi adducts indicate a huge biological difference in their formation and biological activity [1,2,3,4,5,6,7,8,9,10,11,12,13,14,15,16,17,18,19,20,21,22,23,24,25,26,27,28,29,30,31,32,33,34,35,36,37,38,39,40,41,42,43,44].

The lack of information on peptidomimetics such as α-aminoacyl amides is hindering the rational design and further biological application of these derivatives. This is especially important from the aspect of the increasingly common drug resistance of bacteria by lipopolysaccharide (LPS) [45,46,47,48,49,50,51] in hydrophobic parts with newly synthesized compounds. The analyzed *E. coli* bacterial strains have different LPS lengths (strains R2–R4) or no LPS (strain K12). The analysed compounds may, in the future, replace known and commonly used antibiotics, such as bleomycin, streptomycin, kanamycin, or ciprofloxacin [52,53,54]. These antibiotics are now widely used for bacterial infections caused by Gram-negative bacteria (including coliforms, pneumoniae, meningitis, urinary tract, and bone inflammation; Proteus spp., Neisseria spp., *Pasteurella multocida*) and some Gram-positive bacteria. (e.g., *Corynebacterium* spp., *Bacillus anthracis*, *staphylococci*). However, some strains of pathogenic bacteria may develop drug resistance to these antibiotics [52,53,54]. The resistance of microorganisms to these antibiotics may result from the presence of bacterial enzymes that modify and block free -OH and -NH groups responsible for the effects of changes in the amino acid sequence (structure) of the ribosome with which the antibiotic cannot bind. Therefore, it is important to develop new substances similar to β-lactam antibiotics, which will be able to inhibit the biosynthesis of the bacterial cell wall. The bactericidal effect is due to the disruption of the synthesis of bacterial proteins, including those that make up the cell membrane [55,56,57,58,59].

The interaction of imines with isocyanates is mainly focused on the well-known multicomponent Ugi reaction (MCR) [1]. This basic process involves a carboxylic acid group that attacks the intermediate nitrile ion, thus leading, after the Mumma rearrangement, to α-amidoamids. Aminoamids are common in many molecules, such as peptides, proteins, lactams, and many synthetic polymers [59]. However, the direct reaction of imines and isocyanides is less studied in the literature. Reactions between reagents possessing carbonyl, amine, and isocyanide group lead to α-amidoamids.

The new compounds are an alternative as potential substitutes for antibiotics due to their specific structure related to substituents. They can, thus, penetrate the membranes of bacterial cells, damaging the nucleic acid they contain. They usually consist of two functional groups—carbonyl and amines—linked by a single bond between carbon and nitrogen. The biological activity of α-amidoamids is determined by means of a heterocyclic ring connected to a ketone or aldehyde group leading to imine formation via molecular cyclization.

The aim of our research was to check whether the interaction of Ugi reaction products depends on LPS length in *Escherichia coli* K-12 and R2–R4 strains that have different LPS in their outermost layer in the antigen “O” region.

## 2. Materials and Methods

### 2.1. Microorganisms and Media

*E. coli* K-12, R1–R4 strains were received from Prof. Jolanta Łukasiewicz the Ludwik Hirszfeld Institute of Immunology and Experimental Therapy (Polish Academy of Sciences, Warsaw, Poland. Bacteria were cultivated in tryptic soy broth (TSB; Sigma-Aldrich, Saint Louis, MI, USA) liquid medium and on agar plates containing TSB medium. *N*,*N*-Dimethylformamide (DMF) were purchased from Sigma Aldrich (CAS No. 68-12-2, Poznań, Poland), Lanes 1kb-ladder, and Quick Extend DNA ladder, (New England Biolabs, Ipswich, MA, USA).

### 2.2. Experimental Chemistry

NMR analyzes were performed according to the procedure described in the literature [59,60].

### 2.3. General Procedure for Synthesis of Compounds ***1**–**20***

The described individual substrates in the synthesis of α-amidoamides belong to the standard methods of synthesizing this type of compounds and are presented unchanged in many publications of this type [48]. Standard syntheses of compounds from 5a to 5u are methodologically identical to those presented in other publications [58]. Therefore, we present a general formula for their synthesis. In addition, examples of the synthesis of each of the analyzed α-amidoamides are also presented below.

White powder; mp. 78–124 °C; ^1^H NMR (400 MHz, CDCl3) δ ppm = 6.73–7.33 (4–12 H, m, Ph), 4.85–7.02 (1–6 H, m, Ph + CH or CH_2_ or C_3_H_3_), 4.74–6.85 (1 H, s, m, br, NH or CH or C3 H3), 4.41–5.93 (1–2 H, s,t,m, NH or CH or CH_2_), 4.17 (1–3 H, m,s, CH or CH_2_ or 2× CH_3_), 3.66–5.75 (1–6 H, m,s, or CH_3_, 2× CH_3_), 0.85 (1–6 H, m, s, CH_2_ or CH_3_), 1.57–1.95 (1–6 H, s,m CH or CH_2_ or CH_3_ or 2× CH_3_), 0.85–3.62 (1–6 H, m,s, CH or CH_2_ or CH_3_ or 2× CH_3_); ^13^C NMR (100 MHz; CDCl3) δ ppm = 14.16–41.32, 22.37–43.18, 22.71–55.22, 25.14–55.26, 25.30–63.46, 23.10–88.50, 24.56–114.02, 25.17–114.31, 37.10–114.31, 40.78–114.05, 41.26–114.23, 41.62–126.95, 42.54–130.38, 42.84–158.81, 48.55–157.71, 55.28–170.69, 56.35–172.98, 73.03–171.05, 114.05–173.13, 126.02–158.84, 126.98–170.67, 127.39–173.79; HRMS calcd from C_29_H_34_N_2_O_3_Na [M + Na]+: 481.2467 found: 481.2470 to C_30_H_36_N_2_O_4_Na [M + Na]^+^: 511.2573 found: 511.2574.

Yellow powder or Pale yellow; mp. 92–93 °C; ^1^H NMR (400 MHz; CDCl3) δ ppm = 6.82–7.30 (1–15 H, m,s Ph), 6.86 (1–5 H, s,m, Ph, or Ph + CH or NH), 4.52–7.09 (1–4 H, m, Ph), 4.46–7.05 (1–4 H, m,s, NH or CH_2_), 4.19–6.86 (1–4 H, m, NH or CH_2_ or Ph or CH), 3.71–6.86 (1–6 H, s, 2× CH_3_), 0.68–6.79 (1–3 H, m, CH_3_), 1.19–6.72 (1–5 H, m, CH or CH_2_), 1.40–6.33 (1–4 H, m, C_2_H_4_), 1.18–1.38 (1–31 H, s br, m, C_15_H_31_); ^13^C NMR (100 MHz; CDCl_3_) δ ppm = 13.72–22.49, 22.43–30.42, 22.66–41.65, 24.73–42.72, 25.15–55.30, 29.88–114.25, 31.20–114.03, 39.01–114.02, 29.61–114.24, 29.64–128.67, 29.68–128.93, 31.90–129.15, 34.04–130.35, 55.21–128,81, 56.20–128.66, 113.97–129.55, 114.09–158.90, 127.26–129.53, 128.97–134.62, 129.55-, 130.47, 158.79–173.25, 158.87–169.74, 170.81–173.39, 170.61–175.80, 160.45–177.43; HRMS calcd from C_26_H_36_N_2_O_3_Na [M + Na]+: 447.2624 found: 447.2611.to C_32_H_32_N_2_O_4_Na [M + Na]+: 541.2678 found: 541.2671.

#### 2.3.1. Product 5a (AM 93)

White powder; mp. 104–105 °C; ^1^H NMR (400 MHz, CDCl_3_) δ (ppm) = 7.04–7.29 (9 H, m, Ph), 6.61–6.78 (5 H, m, Ph + CH), 4.74 (1 H, s, NH), 4.41 (2 H, s, CH_2_), 4.17 (2 H, m, CH2), 3.66–3.72 (6 H, m, 2× CH_3_), 1.81 (2 H, s, CH2), 1.57 (1 H, s, CH), 0.85 (6 H, s, 2× CH_3_); ^13^C NMR (100 MHz; CDCl_3_) δ ppm = 22.2, 22.7, 25.1, 37.3, 42.7, 51.1, 55.2, 57.9, 113.7, 113.9, 126.7, 128.5, 128.8, 129.0, 129.2, 129.8, 130.4, 136.2, 158.8, 170.7, 173.8; HRMS calcd for C_29_H_34_N_2_O_4_Na [M + Na]+: 497.2416 found: 497.2410.

#### 2.3.2. Product 5b (AM 70)

White powder; mp. 106–107 °C; ^1^H NMR (400 MHz;CDCl_3_) δ (ppm) = 7.19 (4 H, m, Ph), 7.01 (5 H,m, Ph), 6.77 (5 H,m, Ph, CH), 6.71 (1 H, s, CH), 4.94 (1 H, t, NH), 4.44 (2 H, s; CH_2_), 4.16–4.20 (2 H, m, CH_2_), 3.73 (6 H, s, CH_3_), 3.55 (2 H, d, CH_2_), 1.79–1.83 (1 H, m, CH), 1.37–1.43 (2 H, m, CH_2_), 0.75–0.79 (6 H, m, CH_3_); ^13^C NMR (100 MHz; CDCl_3_) δ ppm = 22.4, 22.7, 25.1, 37.0, 41.3, 42.8, 48.2, 55.3, 56.5, 114.0, 114.2, 126.9, 127.3, 128.6, 128.8, 129.0, 129.2, 130.3, 134.6, 158.9, 170.5, 173.4; HRMS calcd for C_30_H_36_N_2_O_4_Na [M + Na]^+^: 511.2573 found: 511.2574.

#### 2.3.3. Product 5c (AM 121)

White powder; mp. 78–79 °C; ^1^H NMR (400 MHz; CDCl_3_) δ (ppm) = 6.74–7.26 (14 H m, PhH + CH), 5.01 (1 H, m, NH), 4.43 (2 H, m, CH_2_) 4.23–4.30 (2 H, m, CH_2_), 3.78 (6 H, s, CH_3_), 2.86–2.91 (2 H, m, CH_2_), 2.53–2.58 (2 H, m, CH_2_), 1.81–1.85 (1 H, m, CH), 1.40–1.44 (2 H, m, CH_2_), 0.81–0.91 (6 H, m, CH_3_); ^13^C NMR (100 MHz; CDCl_3_) δ ppm = 22.4, 22.7, 25.1, 31.4, 35.7, 37.0, 42.8, 48.0, 55.3, 114.0, 114.1, 126.2, 127.2, 128.4, 129.0, 129.3, 130.4, 140.9, 158.8, 170.7, 174.6 ppm; HRMS calcd for C_31_H_38_N_2_O_4_Na [M + Na]^+^: 525.2729 found: 525.2717.

#### 2.3.4. Product 5d (AM 84)

White powder; mp. 123–124 °C; ^1^H NMR (400 MHz; CDCl_3_) δ ppm = 6.73–7.08 (m, 9 H; Ar, CH), 4.85–5.03 (m, 1 H; -NH), 4.44 (s, 2 H, CH_2_), 3.97–4.32 (m, 2 H; CH_2_), 3.71 (s, 6 H; CH_3_), 1.95 (s, 3 H; CH_3_), 1.66–1.84 (m, 1 H, CH), 1.29–1.52 (m, 2 H; CH_2_), 0.78 (t, 6 H; CH_3_);^13^C NMR (100 MHz; CDCl_3_) δ (ppm) = 22.4, 22.7, 25.2, 37.2, 42.8, 48.7, 55.2, 56.1, 114.0, 114.1, 127.3, 129.0, 129.3, 130.4, 158.8, 158.9, 170.7, 173.0; HRMS calcd for C_24_H_32_N_2_O_4_Na [M + Na]+: 435.2260 found: 435.2260.

#### 2.3.5. Product 5e (AM 119)

White powder; mp. 94–95 °C; ^1^H NMR (400 MHz; CDCl_3_) δ (ppm) = 6.84–7.33 (9 H, m, PhH), 6.35 (1 H, m, CH), 4.94–4.97 (1 H, m, NH), 4.50–4.51 (2 H, m, CH_2_), 3.78 (3 H, s, CH_3_), 3.57–3.62 (2 H, m, CH_2_), 0.79–1.82 (20 H, m, C_6_H_11_ + C_4_H_9_); ^13^C NMR (100 MHz; CDCl_3_) δ (ppm) = 22.4, 22.7, 24.7, 25.2, 25.3, 32.7, 32.8, 37.0, 41.4, 47.9, 48.1, 55.3, 56.5, 114.2, 127.0, 127.2, 128.6, 128.7, 129.2, 129.5, 129.9, 134.6, 158.9, 169.6, 173.3; HRMS calcd for C_31_H_38_N_2_O_4_Na [M + Na]+: 525.2729 found: 525.2720.

#### 2.3.6. Product 5f (AM 116)

Yellow powder; mp. 92–93 °C; ^1^H NMR (400 MHz; CDCl*3*) δ (ppm) = 6.95–7.19 (4 H, m, Ph), 6.86 (1 H, s, CH), 6.73–6.77 (4 H, m, PhH), 4.94–4.97 (1 H, m, NH), 4.43–4.44 (2 H, m, CH_2_), 4.19–4.21 (2 H, m, CH_2_), 3.71 (6 H, s, 2× CH_3_), 2.09–2.25 (3 H, m, CH_3_), 1.74–1.80 (1 H, m, CH), 1.39–1.45 (4 H, m, C_2_H_4_), 1.18 (31 H, s br, C_15_H_31_); ^13^C NMR (100 MHz; CDCl_3_) δ (ppm) = 14.1, 22.4, 22.7, 22.8, 29.3, 29.4, 29.5, 29.6, 29.7, 31.9, 34.0, 55.2, 114.0, 114.1, 127.3, 128.0, 129.5, 130.5, 158.8, 158.9, 170.8, 175.8, 177.4; HRMS calcd for C_27_H_30_N_2_O_4_Na [M + Na]+: 469.2103 found: 469. 2092.

#### 2.3.7. Product 5g (AM 115)

White powder; mp. 115–116 °C; ^1^H NMR (400 MHz; CDCl_3_) δ (ppm) = 7.13–7.17 (4 H, m, Ph), 6.97–7.02 (5 H, m, Ph), 6.65–6.77 (5 H, m, PhH + CH) 4.97–4.99 (1 H, s br; NH), 4.42–4.44 (2 H, m, CH_2_), 4.15–4.17 (2 H, m, CH_2_), 3.69 (6 H, s, 2 × CH_3_), 3.52–3.54 (2 H, m, CH_2_), 1.18–1.26 (q, 3 H; CH_3_);^13^C NMR (100 MHz; CDCl_3_) δ (ppm) = 14.2, 42.9, 48.0, 53.4, 55.3, 114.0, 114.3, 127.0, 127.1, 128.7, 129.0, 129.3, 130.4, 134.6, 159.0, 171.0, 173.1; HRMS calcd for C_28_H_32_N_2_O_4_Na [M + Na]+: 483.2260 found: 483.2092.

#### 2.3.8. Product 5h (AM 164)

Pale yellow oil; 1 H NMR (400 MHz; CDCl_3_) δ (ppm) = 7.17–6.96 (9 H, m, PhH), 6.77–6.73 (5 H, m, PhH + CH), 4.83–4.79 (1 H, m, NH), 4.46 (2 H, s, CH_2_), 4.20–4.15 (2 H, m, CH_2_), 3.71 (6 H, s, CH_3_), 3.54–3.53 (2 H, m, CH_2_), 1.89–1.85 (1 H, m, CH), 1.48–1.46 (1 H, m, CH), 1.19–1.17 (2 H, m, CH2), 0.78–0.74 (3 H, m, CH_3_); 13C NMR (100 MHz; CDCl_3_) δ (ppm) = 13.8, 19.7, 30.4, 41.3, 42.8, 48.3, 55.3, 58.2, 114.0, 114.2, 126.9, 127.3, 128.6, 128.8, 129.0, 129.3, 130.3, 134.6, 158.9, 158.9, 170.4, 173.3; HRMS calcd for C29 H34N2O4Na [M + Na]+: 497.2416 found: 475.2416.

#### 2.3.9. Product 5i (AM 165)

Pale yellow oil; ^1^H NMR (400 MHz; CDCl3) δ (ppm) = 7.19–6.93 (10 H, m, PhH), 6.77–6.70 (3 H, m, PhH + CH), 4.57–4.40 (2 H, m, CH2) 4.29–4.24 (2 H, m, CH_2_), 4.14–4.09 (1 H, m, NH), 3.71 (3 H, m, CH_3_), 3.53 (2 H, s, CH_2_), 2.43–2.37 (1 H, m, CH), 0,86 (2 H, m, CH_3_), 0.68–0.68 (3 H, m, CH_3_); ^13^C NMR (100 MHz; CDCl_3_) δ (ppm) = 19.0, 19.9, 27.1, 41.6, 42.7, 55.2, 55.3, 114.0, 114.2, 126.9, 127.7, 128.6, 128.9, 129.1, 130.1, 130.3, 134.6, 158.9, 170.0, 173.7; HRMS calcd for C_29_H_34_N_2_O_4_Na [M + Na]^+^: 497.2416 found: 475.2416.

#### 2.3.10. Product 5j (AM 107)

White powder; mp. 86–87 °C; ^1^H NMR (400 MHz; CDCl_3_) δ (ppm) = 7.07–7.23 (12 H, m, PhH + C_2_H_2_), 6.63–6.85 (6 H, m, C_3_H_3_), 5.93 (1 H, s, NH), 5.75 (1 H, s, CH), 4.62 (1 H, m, CH), 4.40 (1 H, m, CH), 3.67(6 H, 2 s, 2× CH_3_), 3.62 (2 H, m, CH_2_); ^13^C NMR (100 MHz; CDCl_3_) δ (ppm) = 41.3, 43.2, 55.2, 55.3, 63.5, 88.5, 113.7, 113.9, 114.0, 114.1, 126.8, 126. 127.5, 128.5, 128.6, 128.7, 128.9, 129.0, 129.7, 134.7, 134.9, 135.0, 158.6, 158.7, 158.8, 159.7, 169.4, 172.8; HRMS calcd for C_29_H_34_N_2_O_3_Na [M + Na]+: 481.2467 found: 481.2470.

#### 2.3.11. Product 5k (AM 91)

White powder; mp. 82–83 °C; ^1^H NMR (400 MHz; CDCl_3_) δ (ppm) = 7.03–7.34 (12 H, m, PhH), 6.82–6.86 (3 H, m, PhH + CH), 5.06–5.10 (1 H, m, NH), 4.60 (1 H, s, CH), 4.23–4.27 (2 H, m, CH_2_), 3.79 (1 H, s, CH_3_), 3.54–3.65 (2 H, m, CH_2_), 1.86–1.93 (1 H, m, CH), 1.43–1.47 (2 H, m, CH_2_), 0.82–0.91 (6 H, m, 2× CH_3_); ^13^C NMR (100 MHz; CDCl_3_) δ (ppm) = 21.7, 22.4, 22.8, 23.1, 24.6, 25.2, 37.1, 40.8, 41.3, 41.6, 42.5, 42.8, 48.5, 55.3, 56.4, 73.0, 114.0, 126.0, 127.0, 127.4, 128.7, 128.8, 129.0, 130.3, 134.5, 137.5, 158.91, 170.4, 173.5; HRMS calcd for C_28_H_40_N_2_O_3_Na [M + Na]+: 475.2937 found: 475.2934.

#### 2.3.12. Product 5l (AM 163)

Pale yellow oil; ^1^H NMR (400 MHz; CDCl_3_) δ (ppm) = 7.20–7.16 (3 H, m, PhH), 7.10–7.08 (2 H, m, PhH), 6.88–6.85 (1 H, m, CH),4.87–4.83 (1 H, m, NH), 4.27–4.13 (2 H, m, CH), 3.71 (3 H, s, CH_3_), 3.62 (2 H, s, CH_2_), 3.15–3.10 (2 H, m, CH_2_),1.80–1.73 (1 H, m, CH), 1.59–1.52 (1 H, m, CH), 1.40–1.47 (2 H, m, CH_2_), 1.18–1.13 (4 H, m, C_2_H_4_), 0.85–0.79 (10 H, m, CH_3_ + CH); ^13^C NMR (100MHz; CDCl_3_) δ (ppm) = 13.6, 20.3, 22.4, 22.8, 24.8, 32.0, 36.7, 41.1, 42.8, 45.6, 55.2, 55.9, 113.9, 126.9, 128.7, 128.7, 129.0, 129.3, 130.5, 134.8, 158.8, 171.4, 172.8; HRMS calcd for C_26_H_36_N_2_O_3_Na [M + Na]^+^: 447.2624 found: 447.2624.

#### 2.3.13. Product 5m (AM 162)

Pale yellow oil; ^1^H NMR (400 MHz;CDCl_3_) δ (ppm) = 7.19–7.17 (3 H, m, PhH), 7.09–7.05 (4 H, m, PhH), 6.74–6.74 (2 H, m, PhH), 4.34–4.17 (2 H, m, CH), 3.96–4.00 (1 H, m, NH), 3.71 (3 H, m, CH_3_), 3.65 (2 H, s, CH_2_), 2.34–2.29 (1 H, m, CH), 1.58–1.52 (2 H, m, CH_2_), 1.18–1.14 (1 H, m, CH_2_) 1.02–1.00 (6 H, m, CH_3_), 0.89–0.88 (6 H, m, CH_3_); ^13^C NMR (100 MHz; CDCl3) δ (ppm) = 20.5, 21.0, 22.3, 25.5, 39.6, 42.6, 42.9, 50.7, 55.2, 59.5, 113.9, 126.9, 128.4, 128.8, 128.8, 129.5, 130.9, 134.8, 158.7, 172.5, 172.8; HRMS calcd for C_26_H_36_N_2_O_3_Na [M + Na]+: 447.2624 found: 447.2618.

#### 2.3.14. Product 5n (AM 182)

Pale yellow oil; ^1^H NMR (400 MHz; CDCl_3_) δ (ppm) = 7.19–7.14 (4 H, m, PhH), 7.04–6.96 (5 H, m, PhH), 6.83–6.81 (1 H, m, CH), 6.74–6.72 (2 H, m, PhH), 6.37–6.33 (2 H, m, PhH), 5.00–4.96 (1 H, s br, NH), 4.42 (2 H, s, CH_2_), 4.22–4.20 (2 H, m, CH_2_), 3.74–3.69 (9 H, m, 3 × CH_3_), 3.51–3.50 (2 H, m, CH_2_), 1.76–1.72 (1 H, m, CH), 1.39–1.35 (2 H, m, CH2), 0.77–0.73 (6 H, m, 2× CH_3_); ^13^C NMR (100 MHz; CDCl_3_) δ (ppm) = 22.5, 22.7, 25.0, 37.0, 38.9, 41.3, 47.8, 55.2, 55.3, 55.4, 98.5, 103.3, 114.2, 118.7, 126.8, 127.1, 128.6, 128.8, 129.5, 130.3, 134.7, 158.6, 158.8, 160.4, 170.2, 173.2; HRMS calcd for C_31_H_38_N_2_O_5_Na [M + Na]+: 541.2678 found: 541.2671.

#### 2.3.15. Product 5o (AM 111)

Pale yellow oil; ^1^H NMR (400 MHz;CDCl_3_) δ ppm = 7.15 (5 H, m, PhH), 6.72–7.03 (5 H, m, PhH + CH), 4.74–5.00 (1 H, m, NH), 4.04–4.32 (2 H, m, CH_2_), 3.71 (3 H, m, CH_3_), 3.55–3.65 (2 H, m, CH_2_), 2.95–3.26 (2 H, m, CH_2_), 1.06–1.89 (12 H, m, C_10_H_10_, CH_2_), 0.81 (9 H, m, CH_3_); ^13^C NMR (100 MHz; CDCl_3_) δ (ppm) = 13.8, 22.3, 22.4, 22.7, 24.7, 26.7, 29.9, 31.2, 36.7, 41.0, 42.7, 45.7, 55.1, 55.8, 98.5, 103.1, 114.5, 118.9, 126.8, 127.6, 128.7, 128.8, 129.5, 130.3, 134.7, 158.6, 158.8, 160.4, 170.2, 173.2; HRMS calcd for C_32_H_32_N_2_O_4_Na [M + Na]: 475.2934 found: 475.2937.

#### 2.3.16. Product 5p (AM 114)

Pale yellow oil; ^1^H NMR (400 MHz;CDCl_3_) δ (ppm) = 6.82–7.30 (15 H, m, PhH + CH), 5.02–4.99 (1 H, m, NH), 4.52 (2 H, s, CH_2_), 4.30–4.35 (2 H, m, CH_2_), 3.80 (3 H, s, CH_3_), 3.61–3.63 (2 H, m, CH_2_), 1.87–1.91 (1 H, m, CH), 1.45–1.49 (2 H, m, CH2), 0.82–0.86 (6 H, m, 2 CH_3_); ^13^C NMR (100 MHz; CDCl_3_) δ (ppm) = 22.4, 22.7, 25.2, 43.4, 55.3, 114.2, 127.3, 127.7, 128.6, 128.7, 145.5, 158.9, 170.7; HRMS calcd for C_28_H_32_N_2_O_2_Na [M + Na]+: 451.2361 found: 451.2358

#### 2.3.17. Product 5r (AM 170)

Pale yellow oil; ^1^H NMR (400 MHz; CDCl_3_) δ (ppm) = 7.24–7.15 (4 H, m, PhH), 7.11–7.10 (2 H, m, PhH), 7.05–7.03 (2 H, m, PhH), 6.86 (1 H, s, CH), 6.81–6.79 (2 H, m, CH_2_), 5.00–4.97 (1 H, s br, NH), 4.46 (2 H, s, CH_2_), 4.15–4.10 (2 H, s, CH_2_), 3.83–3.80 (2 H, m, CH_2_), 3.73 (3 H, s, CH_3_), 3.62 (2 H, m, CH_2_), 1.82–1.78 (1 H, m, CH), 1.43–1.36 (2 H, m, CH_2_), 1.22–1.18 (3 H, m, CH_3_), 0.79–0.75 (6 H, m, 2× CH_3_); ^13^C NMR (100 MHz; CDCl_3_) δ (ppm) = 14.1, 22.3, 22.7, 25.1 36.8, 41.1, 41.2, 48.3, 55.3, 56.2, 61.3, 114.20, 126.7, 127.3, 128.6, 128.8, 129.7, 134.6, 158.8, 169.7, 173.5; HRMS calcd for C_26_H_34_N_2_O_5_Na [M + Na]^+^: 477.2365 found: 477.2362

#### 2.3.18. Product 15s (AM 178)

Pale yellow oil; ^1^H NMR (400 MHz;CDCl_3_) δ (ppm) = 7.24–7.15 (4 H, m, PhH), 7.11–7.10 (2 H, m, PhH), 7.05–7.03 (2 H, m, PhH), 6.86 (1 H, s, CH), 6.81–6.79 (2 H, m, CH_2_), 5.00–4.97 (1 H, s br, NH), 4.46 (2 H, s, CH_2_), 4.15–4.10 (2 H, s, CH_2_), 3.83–3.80 (2 H, m, CH_2_), 3.73 (3 H, s, CH_3_), 3.62 (2 H, m, CH_2_), 1.82–1.78 (1 H, m, CH), 1.43–1.36 (2 H, m, CH_2_), 1.22–1.18 (3 H, m, CH_3_), 0.79–0.75 (6 H, m, 2× CH_3_); ^13^C NMR (100 MHz; CDCl3) δ (ppm) = 14.1, 22.3, 22.7, 25.1 36.8, 41.1, 41.2, 48.3, 55.3, 56.2, 61.3, 114.20, 126.7, 127.1, 128.6, 128.6, 129.5, 134.6, 158.9, 169.7, 173.2; HRMS calcd for C_28_H_40_N_2_O_3_Na [M + Na]+: 475.2937 found: 475.2932.

#### 2.3.19. Product 5t (AM 171)

Pale yellow oil; ^1^H NMR (400 MHz; CDCl_3_) δ ppm = 7.22–7.19 (4 H, m, PhH), 7.09–7.01 (3 H, m, PhH), 6.82–6.80 (2 H, m, PhH), 6.35 (1 H, s, CH), 4.91 (1 H, s br, NH), 4.45 (2 H, s, CH_2_), 3.74 (3 H, s, CH_3_), 3.58–3.56 (2 H, m, CH_2_), 3.08–3.06 (2 H, m, CH_2_), 1.80–1.76 (1 H, m, CH), 1.34–1.22 (6 H, m, C_3_H_6_), 0.85–0.74 (9 H, m, 3× CH_3_); ^13^C NMR (100 MHz; CDCl_3_) δ (ppm) = 13.7, 20.0, 22.4, 22.8, 25.1, 31.5, 37.0, 39.0, 41.3, 48.1, 55.3, 114.2, 127.0, 127.2, 128.7, 128.8, 129.4, 134.6, 158.9, 170.6, 173,4; HRMS calcd for C_26_H_36_N_2_O_3_Na [M + Na]+: 447.2624 found: 447.2611.

#### 2.3.20. Product 5u (AM 184)

Pale yellow oil; ^1^H NMR (400 MHz; CDCl_3_) δ (ppm) = 7.29–7.09 (5 H, m, PhH), 7.04–7.02 (2 H, m, PhH), 6.82–6.77 (2 H, m, PhH), 6.18 (1 H, s, CH), 4.87–4.83 (1 H, s br, NH), 4.44–4.43 (2 H, m, CH_2_), 3.73 (3 H, s, CH_3_), 3.57–3.56 (2 H, m, CH_2_), 1.78–1.69 (1 H, m, CH), 1.38–1.17 (10 H, m, 3× CH_3_ + CH), 0.87–0.74 (8 H, m, 2× CH_3_ + CH_2_); ^13^C NMR (100 MHz; CDCl_3_) δ (ppm) = 22.4, 22.7, 25.2, 28.6, 36.9, 41.4, 47.7, 51.0, 55.3, 56.8, 114.2, 127.0, 127.2, 128.6, 128.7, 129.6, 134.7, 158.9, 169.7, 173.2; HRMS calcd for C_26_H_36_N_2_O_3_Na [M + Na]^+^: 447.2624 found: 447.2614.

### 2.4. Determination of MIC and MBC

Determination of MIC and MBC for selected strains are described in detail in the literature [31,45,51]. (An example of the analysis are presented in Appendix A.)

#### 2.4.1. Interaction of Bleomycine, Kanamycine, Streptomycine, with Bacterial Plasmid DNA Isolated from K12 and R1–R4 Strains

Plasmid DNA was isolated from the used bacterial influences using Labjot kits according to the manufacturer’s protocols. Next, the obtained DNA plasmid was digested with antibiotics such as bleomycin with final working concentration (2 units/mL), streptomycin (50 µg/mL), ciprofloxacin (10 µg/mL), and kanamycin (50 µg/mL) as per the manufacturer’s instructions. All antibiotics were purchased from Sigma (Poznań, Poland).

#### 2.4.2. Determination of MIC and MBC after Antibiotics Treatment

Analyzed strains will be treated with antibiotics at identical concentrations as the analyzed compounds using the MIC and MBC tests according to the procedures described in previous publications [31,45,51].

### 2.5. Interaction of the Plasmid DNA from K12 and R4 Strains with α-Amidoamids

Strains K12 and R4 were selected for further studies on the basis of the MIC (Minimum inhibitory concentration) and MBC (Minimum bactericidal concentration) analysis. The method of interaction of plasmid DNA with the analyzed α-amidoamides has been described in detail in the literature [48].

### 2.6. Repair and Cleavage of Oxidative DNA Damage Adducts by Fpg Protein in Bacterial Cells

Analysis of repair and cleavage by Fpg protein were performed on (i) plasmid DNA isolated from different *E. coli* mutants of R1–R4 and K12 by alkaline lysis and final purification according to the manufacturer’s protocol (Labjot, Warsaw, Poland). The plasmids were then incubated with the analysed compounds.

Subsequently, DNA was digested with Fpg enzymes individually as described in detail in the literature [50].

### 2.7. Statistical Analysis

The obtained data were used in the statistical analysis (Statistica package ver. 5.0) and presented as standard error of the mean (SE). Parametric analyzes were performed using Student’s *t*-test *p* < 0.05 *, *p* < 0.01 ** and *p* < 0.001 ***.

## 3. Results

### 3.1. Chemistry

Target α-amidoamids 5a–5u were obtained in Ugi reaction. The reaction between carboxylic acids, aldehydes, amines, and isocyanides was performed in the presence of vesicles in an aqueous medium (Figure 2).

Application of a surfactant for reaction leads to the formation of vesicles promoting reaction progress. Thus, toxic transition metals were excluded from the reaction medium. The presence of transition metal traces is intolerable for pharmaceutical and cosmetic products and should be avoided. The individual α-amidoamides obtained by the Ugi reaction and isolated *via* column chromatography are shown below in Figure 3. 

### 3.2. Toxicity of Tested Compounds

MIC and MBC are frequently used assays to determine the cytotoxic activity of chemotherapeutic compounds. These simple tests are based on the examination of the integrity of the bacterial cell membrane, since any permanent damage to the cell membrane leads to cell death and changes the color of the analyzed sample. The tested bacterial cells are incubated with a positively charged dye solution. The dye used does not penetrate living cells, but dying cells (when the membrane is permanently damaged and the potential between the outer and inner side of the membrane is lost), staining the cytoplasm and/or the genetic material.

In this study, we paid attention to the determination of the biological activity of peptidomimetics (Figure 4, Figure 5 and Figure 6). Experiments were performed with Ugi reaction products 5 obtained from various carboxylic acids, aldehydes, and isocyanates. To test the toxicity of the analyzed compounds, the MIC and MBC tests were selected. The differences in the MIC and MBC values for each approximate of all *E. coli* strains and K12 and all analyzed compounds are presented in Figure 4, Figure 5 and Figure 6 as well as Appendix A.

First, the influence of carboxylic acid R_1_ group was investigated. The product of reaction between *p*-methoxybenzylamine, isovaleraldehyd, *p*-methoxybenzyl isocyanide and various carboxylic acid such as benzoic acid, phenylacetic acid, phenyl propanoic acid, acetic acid and α-methoxyphenylacetic acid were investigated (**5a**–**5e**, respectively). The best values of MIC and MBC resulted in compounds **5a** and **5e** (about 0.8 µg/mL for MIC and 15 µg/mL for MBC). Then, biological activity of compounds **5f**–**5i** derived from various aldehydes such as ethanal, propanal, butanal, 2-methylpropanal were investigated. Substantial influence of the R_2_ group derived from aldehydes was noted. The most reactive was product **5g** received in the reaction with phenylacetic acid, *p*-methoxybenzylamine, *p*-methoxybenzyl isocyanide, and propanal (about 0.8 µg/mL and 15 µg/mL for MBC). Then, the effect of amine R_3_ group change on the biological activity of peptidomimetics **5j**–**5m** was determined. Compounds 5j possessing benzyl group and **5k** possessing hexyl group gave the best results (about 0.8 µg/mL for MIC and 15 µg/mL). In order to determine the effect of the R_4_ group on the activity of target compounds **5**, the Ugi reaction was performed with different isocyanides providing compounds **5n**–**5u**. The most reactive were compounds **5n**, **5t,** and **5u** derived from 2,5-dimethoxybenzyl isocyanide, *n*-butylizocyanide, and *tert*-butyl isocyanide, respectively, (about 1.2 µg/mL for MIC and 30–35 µg/mL for MBC). Compounds **5p** possessing cyclohexyl isocyanide gave high values of MIC and MBC as well (about 0.8 µg/mL and 30 µg/mL for MBC).

On the first plate (Panel A) where the strains R4 were sequentially added together with the 20 analyzed compounds (Appendix A), the color change was visualized at a 10^−3^ dilution, which corresponded to a MIC value of 0.02 µg mL^−1^. Among all tested compounds, especially compounds with *n*-butylizocyanide **5t**, 2,5-dimethoxybeznyl isocyanide **5n** and *tert*-butyl isocyanide **5u** gave a visible color. On the second plate (panel B), where the strains R3 were sequentially added together with all 20 analyzed compounds (Appendix A), a color change was analysed also at a dilution of 10^−3^ and a MIC value of 0.02 mg mL^−1^ by compounds designated as **5a**, **5b**, **5e**, **5g**, **5j**, **5k**, **5p**, and **5o** gave a visible color. A similar effect was observed on the third plate (panel C) where the strains R2 were used. Compounds **5a**–**5g**, **5j**, **5k**, **5p**, and **5o** (Appendix A) gave a visible color. On the fourth plate, where the strains K12 were sequentially added together with the 20 analyzed compounds (Appendix A), a color change was observed, also at a dilution of 10^−2^ and a MIC value of 0.001 µg mL^−1^ by compounds designated as **5a**–**5u**, giving a visible color. The interactions between the analyzed strains and compounds are presented in Figure 4. Increasing MIC values were observed for all analyzed compounds, but the most reactive were compounds **5a**, **5e**, **5g**, **5j**, **5k**, **5n**, **5p**, **5t**, and **5u**. All bacterial strains R2, R3, and R4 showed various (but higher) sensitivity to the analyzed compounds than the K12 strain. The highest activity was observed for the strain R4 > R3 > R2 (Figure 4). The R4 strain was probably the most sensitive than other strains (Figure 4 and Figure 5). In 20 analyzed cases using 96-well plates, the observed MBC/MIC values were approximately 75 times higher than the MIC (Figure 6) and were statistically significant, as shown in Table 1.

### 3.3. Modification of Plasmid DNA Isolated from E. coli R2–R4 Strains with Tested α-Amidoamids

The increase in toxicity of all the analyzed compounds (high MIC values) depends on the functional groups R_1_, R_2_, R3, and R_4_ of α-amidoamides **5** used. We expected a similar effect in plasmid DNA modified with α-amidoamides, in which plasmid damage should be most visible. The tested compounds showed very strong disturbances in the structure and conformation of DNA, even when digesting the Fpg protein in vitro during the 24-h experiment.

The analysis of the MIC and MBC values obtained from all the analyzed strains (K12 and R2–R4) prompted us to isolate plasmid DNA from them and treat them with Fpg protein from the group of repair glycosases. Fpg protein digestion of the obtained plasmids isolated from both the control (K12) and the analyzed (R2–R4) strain after treatment with α-amidamides, regardless of different functional groups, showed clearly visible damage in the topological changes of plasmid DNA forms; covalently closed circle (ccc), linear form, open form (oc), and fuzzy bands (see Appendix A).

For unmodified plasmids from strains K12 and R2-R4, three traditional forms were observed: very weak oc, linear, and ccc. In the modified plasmids, we see significant changes between the control and the α-amidoamide-modified plasmids on electrophoretic images (Appendix A). In the drawings, Figure 7 and Appendix A, it was shown that all analyzed compounds with different substituents strongly changed the topological forms of plasmids, even after digestion with Fpg enzyme.

In plasmid DNA isolated from all strains, mainly the α form was observed. About 3% or more of the oxidative damage was identified after digestion with Fpg, which may indicate that α-amidoamides containing various functional groups in their structure, including isocyanates, can damage plasmid DNA and may be potential new substrates for this protein. It was also found that the interaction with the genetic material of the analyzed compounds is highly toxic for the bacterial cell (change in the composition and length of the bacterial LPS), which leads to the induction of oxidative stress in bacterial cells, where plasmid DNA forms are visibly degraded.

A significant influence on the reactivity of the analyzed compounds has steric factor and structural properties of the different isocyanides of all analysed peptidomimetics such as: *tetr*-butyl isocyanide or 2,5-dimethoxybenzyl isocyanide, which determine their toxicity for specific *E. coli* R-type strains (after observation values MIC and MBC at the concentrations of 0.02 mg (dilutions of 10^−3^ mL) or 0.002 mg (dilutions 10^−4^ mL) in all analyzed strains.

Therefore, the next step in our research was the use of peptidomimetics as potential analogues of well-known antibiotics. The experimental system was identical to the α-amidamide derivatives **5** used with the use of MIC and MBC tests, both in terms of quantity and concentration (Figure 8). In all analyzed R-type strains, a color change was observed in the wells for all tested antibiotics at a dilution of 10^−3^, which corresponds to 0.02 mg mL^−1^ in the analyzed MIC (Appendix A). The greatest effect of the analyzed MIC for antibiotics was observed for bleomycin in strain R3, followed by ciprofloxacin in strain R3 and R4, streptomycin in strain R4, and kanamycin in strain R4 (Figure 8). The R4 strain, having the longest LPS chain, has been shown to interact with all active groups contained in antibiotics. The results were statistically significant at <0.05.

Based on the effect of α-amidoamides with the Fpg protein, a similar experimental system was used for the analyzed antibiotics in terms of dose and concentration (Figure 9, Appendix A).

In plasmid DNA isolated from both strains, and “reacted” with a set of four antibiotics, the oc form was mainly observed (Appendix A). After antibiotic treatment, the level of DNA damage were not on a similar level (Figure 9).

The highest values of observed damage in plasmid DNA were observed for the antibiotics bleomycin > kanamycin > streptomycin > cirpofloxacin in all the analyzed strains. The strains R4 > R2 > R3 > K12 showed the greatest sensitivity to reactions with antibiotics. The values of detected base damage in plasmid DNA after digestion with Fpg protein and bleomycin treatment were 2-fold higher compared to other antibiotics in all the analyzed strains. The values for ciprofloxacin, kanamycin and streptomycin were similar for the strains K12, R2, and R3, but for the strain R4, they reached 1.5 times the values after treatment with kanamycin.

The microbiological toxicity of targeted antibiotics in the case of α-amidoamides significantly affects the structure and length of the LPS of bacteria.

The microbiological activity of the applied antibiotics after digestion with the Fpg protein was the highest in the R4 strain, then R3 and R2, and finally in the K12 strain. Digestion of the modified plasmids with the Fpg protein increased the intensity of the bands in all plasmid DNAs isolated from the strains.

In unmodified DNA plasmids isolated from bacteria, no differences in the analyzed forms were observed. Only two forms of plasmids have been distinguished; the oc form and the broad band migrating between the linear and ccc forms on an agarose gel (Appendix A). Since the endogenous level of Fpg glycosylase in unmodified bacteria is very low, it is possible that all guanine residues have not been fully repaired in the plasmid DNA and may interfere with one or more base excision repair system (BER)-related bacterial repair enzymes or topoisomerases. This suggests that residues persist for more than 24 h in unmodified bacteria. Stabilization of the topoisomerase cleavage complex is essential for the cell as it blocks replication and transcription. Moreover, the secondary effect of 8oxoG persistence in the genome influences the global super-helical DNA density.

## 4. Discussion

Determination of the cytotoxicity of chemotherapeutic compounds to bacterial cells in vitro is an important introduction to the generally understood toxicological studies necessary to determine the correct operation and safety of the drug. The usage of cell models in toxicological studies has many advantages, such as speed and ease of studying cellular and molecular processes, repeatability, the possibility of using small amounts of test substances, and the ability to work on human cells.

The antibacterial effectiveness of the compounds used was high for **5a**, **5e**, **5g**, **5j**, **5c**, **5s**, and the highest for **5n**, **5t**, and **5u**. All data were statistically significant. The reduction in antimicrobial efficacy was observed for **5a**, **5h**–**5n**, **5p**, and **5r**. Antimicrobial efficacy was low with the other compounds. This means that the type of substituent and the type of compound are critical for its toxicity in interaction with bacterial cells.

In the analyzed α-amidoamides **5** containing various R_4_ groups in their structure, toxicity may increase in relation to the subsequent strains of *E. coli* R2 > R3 > R4 differing in LPS length. It is known from the literature that the analyzed strains may cause malfunction of the circulatory and gastrointestinal systems, often leading to cancer [31,45]. The potential toxicity of all α-amidoamides **5** for all bacterial cells was high, especially in strains R2 and R4 on K12, R2 and R3 [16,21,22,31,45,51], for compounds containing R_4_ group as a short chain alkyls and derived from various isocyanides such as isocyanates *tert*-butyl compound **5u** or 2,5-dimethoxybenzyl isocyanate compound **5n** or n-butylisocyanate compound **5t** (Figure 3); MBC values were higher than the MIC (Figure 4) for the analyzed strains [48]. The α-amidoamids **5** with different R_4_ groups were more effective for the K12 and R2–R3 strains. The effect was also similar after interaction of the bacterial membrane with ionic liquids with the quaternary ammonium surfactants [45,51].

Changes in the rearrangement of the polarity of the LPS bacterial membrane components as a result of interaction with α-amidoamides show a strong toxicity (oxidative stress) for the bacterial cell, which may lead to its biochemical decomposition [6,12] and damage to its genetic material. The obtained results constitute the basis for the continuation of further research on other pathogenic bacterial strains associated with diseases of the digestive, cardiovascular, respiratory, genitourinary systems and oral microbiota, and will be necessary to determine the potential mechanisms of their degradation in cell membranes, through new synthesized substances as new antibiotic precursors.

Analysis of the toxicity of α-amidoamids **5** used in this research shows that it is strongly related to the length of the LPS in the analyzed types of bacteria R2–R4. Our research were estimated by a comparison of samples digested or not by *N*-glycosylase/AP lyase DNA [61].

The percentage of damage to the plasmids modified by α-amidamides **5** was determined on the basis of changes in topological forms after digestion with Fpg glycosylase. From the literature data it is known that the Fpg protein has two activities, glycosylase and A-lase. Fpg glycosylase has a broad spectrum to recognize and eliminate oxidized and alkylated bases that have been modified by ROS or RNS. The Fpg protein is now recognized by many world laboratories as an extremely sensitive “marker” of oxidized bases generated in bacterial cells under the influence of oxidative stress induced by internal and external factors. The amount of identified oxidative or alkylation damage to DNA bases over 3–4% by this protein is a very important indicator of the degree and strength of modified guanines or adenines in the analyzed genetic material [32,61].

In our research, the protein-Fpg selectively recognizes modifications introduced by α-amidoamides **5** to plasmid DNA, especially for compounds **5s**, **5t** and **5u**. Visible changes among the three bacterial topological forms of plasmid DNA—the so-called “smear” of bands after digestion with repair glycosylase.

The results suggest that all peptidomimetics tested modify plasmid DNA, which is recognized by Fpg glycosylase (see Figure 3, Figure 7 and Appendix A). Compounds **5a**–**5e**, **5g**, **5j**, **5n**, **5s**, **5t** show the most effective results. This means that in the future, specific α-amidoamides **5** marked with symbols **5c**–**5e**, **5g**, **5j** can be designed as new substitutes for antibiotics with a very similar chemical structure.

According to the literature, frequent chemotherapy with antibiotics, such as ciprofloxacin, kanamycin, streptomycin, and bleomycin resulted in immunization of many pathogenic bacteria [60,62,63,64,65,66,67,68,69,70,71,72,73,74,75,76,77,78]. After treating the plasmid DNA with these antibiotics and digested Fpg protein, the highest level of damage was observed for the strain R4 > R2 > R3 > K12 after bleomycin treatment and were 2-fold higher compared to other antibiotics in all analyzed strains (Figure 9).

The toxicity values for the analyzed ciprofloxacin, kanamycin, and streptomycin were at a similar level, but only for the K12, R2, and R3 strains. For the R4 strain, a 1.5-fold increase in value was observed after treatment with kanamycin compared to the other antibiotics, ciprofloxacin, and streptomycin.

Therefore, it is justified to search for compounds with a similar or greater toxicity to bacterial cells, which are similar in structure and function to the antibiotics used.

Based on the analysis of our research results using the MIC and MBC tests, we can design α-amidoamides **5** with very high toxicity to gram negative bacterial cells. The practical application of the α-amidoamides analyzed by us will enable them to be used in the future as new “antibiotics” that are more toxic and effective than those currently used.

In their biological activity, they are similar to aminoglycosides and β-lactams. At higher concentrations, similar to aminoglycosides and β-lactams, they destroy the complex of ribosomes with mRNA, which inhibits protein synthesis in the bacterial cell [54], and also inhibits bacterial DNA topoisomerase and DNA gyrase [52].

The Ugi multicomponent reaction is employed for drug discovery. Research has shown that the water molecule formed as a by-product accelerates the reaction. The reaction is extremely useful for easy synthesis libraries of compounds. In DNA isolated from the control K12 strain of *E. coli,* we observed any changes between the ratio of closed circular (ccc) to (on) forms of the unmodified and modified bacterial DNA after treatment by peptidomimetics on cleavage with Fpg enzymes in different combinations (Appendix A). In all analyzed DNA plasmids, only the ccc form was observed, while the oc and linear forms were lost. In the **5n** and **5u** samples incubated with the analyzed compounds and digested with Fpg glycosylase, we observed a “large smear” and a change in the ccc form ratio. This means that the enzymes recognized any oxidized damage that was more “toxic” to DNA than other compounds in this group. In panel B, we observed very strong fringes of the ccc form on all the pathways digested by the Fpg enzymes, also observed as “smear”. After the enzyme was incubated with all the modified plasmids, no changes in topological form were observed during electrophoresis in plasmid DNA isolated from *E. coli* strain R4.

Only one band migrated with significant retardation, suggesting the formation of a high molecular weight complex of DNA with the protein. No change in mobility and separation of the topological forms was observed when unmodified plasmid was incubated with Fpg prior to electrophoresis. The Fpg protein by β-elimination removes modified oxidized and alkylated bases elimination; and (ii) displays a dRPase activity [60,62,63,64,65,66,67,68,69,70,71,72,73,74,75,76,77,78]. The structure of Fpg comprises of two domains. The *N*-terminal domain contains the active site within the first 72 amino acid residues and the secondary amino group of the conserved Pro1. Pro1 and Glu2 are indispensable to glycosylase activity. Lys56 and Lys154 are involved in substrate recognition. The enzyme recognizes oxidized purines via the C8 keto group or the carbonyl moiety. The C-terminal domain contains the helix-harpin-helix (HhH) motif and participates in DNA binding. Other functional homologs of formamidopyrimidine DNA glycosylase have been identified in *Saccharomyces cerevisiae* (yOGG1) and humans (hOGG1). The yeast and human OGG1 proteins possess a DNA glycosylase/AP-lyase activity that releases 8-oxoguanine and FapyG [60,62,63,64,65,66,67,68,69,70,71,72,73,74,75,76,77,78].

In strain R4, cleavage of the modified plasmid from Fpg resulted in the disappearance of the CCC form and the appearance of a single band that migrated similarly but slightly slower than the OC form in the modified plasmid, forming a high molecular weight complex (see, Appendix A). This may suggest that Fpg forms a strong complex with certain modified DNA solutions and that the complex is stable under electrophoresis conditions. The protein was covalently bound to DNA, and after digestion with Fpg, amino acids or protein fragments still remained bound to the DNA bases. This suggests that the Fpg protein recognizes some induced DNA damage following modification with **5n** and **5u** compounds. The Fpg protein is also able to bind to the modified DNA, and the use of the enzyme increases the potential formation of a DNA-high molecular weight protein complex.

This suggests that protein-DNA crosslinking via aldehyde adduct groups may be a ubiquitous mechanism; however, the extent of this may be different for different proteins. The repair of the analyzed three forms of DNA plasmids formed during incubation by four different antibiotics is carried out at least through the BER pathways. In our study, we found that the system in *E. coli* is involved in processing replication forks similar to those blocked by ethene DNA adducts after application of the repair glycosylase-Fpg. This enzyme is one of the repair systems by excision of some of the bases involved in the removal of modified DNA-DNA base adducts from the template. We also observed that the modified plasmid was digested with the Fpg protein used as AP lyase. The enzyme from the BER pathway studied in this study—Fpg glycosylase—forms DNA-protein complexes that migrate more slowly during agrose gel electrophoresis.

We postulate that the interaction between DNA, α-amidoamides **5** and antibiotics involved a covalent bond between the aldehyde group and the R_4_ group of target compounds derived from isocyanides that can arise in single-stranded plasmid DNA. Each band represents the nucleotide position at which strand damage or break was induced by digestion with DNA glycosylase/AP-lyase. Analysis of the mapped enzymatic recognition sites suggests that all DNA residues are linked to modified oxidized bases that may correspond to the 8-oxoguanine sites in the template. Probably the base pair substituted with a heptyl side chain residue rearranged to lose the side chain is efficiently cleaved by the Fpg glycosylase and is a novel substrate for them.

Base modifications were also detected at sequences that caused steric hindrance for DNA polymerase on an unmodified template, as “smears’ upon modification of DNA. Addition of specific groups of α-amidoamids cause thermodynamic destabilization of the single helix on cytosine and guanine [78], and relax the compact structure of CG-rich regions, which were bypassed more readily by Fpg enzymes, allowing identification of damaged cytosine residues. The nature of formation of DNA derivatives recognized by Fpg glycosylase is not clear, and needs to be further clarified.

In the Fpg protein, two lysine residues, Lys-57 and Lys-155, are involved in catalysis and directly interact with the C8 of 8-oxopurines [73].

In addition, Fpg protein Pro2 is engaged in N-glycosidic bond breakage by forming a Schiff base with C1 of deoxyribose. This suggests that the N-terminal amino group is very likely to be available for binding specific groups in the components and antibiotics [60,62,63,64,65,66,67,68,69,70,71,72,73,74,75,76,77,78].

In general, the R2-R4 and K12 strains displayed higher sensitivity than the R4 strain (Figure 3, Figure 4, Figure 5 and Figure 6, Appendix A).

Our research shows that both the assays used, MIC and MBC, were successfully applied to all peptidomimetics of structure **5** (Figure 4, Figure 5 and Figure 6). In general, the analyzed LPS-containing bacterial strains showed an increased, with different severity (compared to the control strain K12), sensitivity to the toxic effects of all α-amidoamides. The R4 strain was probably the most sensitive and was very similar in terms of sensitivity to the K12 strain. In all analyzed cases, the MBC value was approximately four times higher than that of the MIC (Figure 6).

The conducted research on the biological activity of all obtained compounds will allow for a better selection of the type and kind of substituents leading to the derivatives with the highest biological activity and the most appropriate tissue properties for the mentioned bacterial cells.

## 5. Conclusions

The present study validates the utility of massive screening for inhibitors of bacterial-specific DNA damage to expedite the discovery of antibiotics with novel modes of action. We showed that: -α-amidoamids of structure **5** are able to modify all *E. coli* strains (R2–R4) and their plasmid DNA, changing the spatial structure of LPS contained in their cell membrane.

-The R4-type strain was the most sensitive among the tested *E. coli* strains.-The insertion of analysed compounds into the leaflet of the outer membrane of the *E. coli* K-12 and rough strains showed that differences in the O-antigen and truncated oligosaccharide core may play important roles in the cellular response to alpha-amidoamids.-The toxicity of alkyl groups depends on their interaction with the membrane, which can incorporate into cell wall structures and change their hydrophobicity.-Membrane rearrangements and disruption may, in turn, result in changes of bacterial responses to other biologically active compounds such as antibiotics [79].-Plasmid DNA damage has been associated with the structure of verified peptidomimetics, suggesting that the presence of R_4_
*tetr*-butyl or 2,5-dimethoxybenzyl groups influences bacterial LPS and generates oxidative stress, which was already observed in our previous studies [50].-The tested α-amidoamides 5 show a different influence on the MIC, which is strongly correlated with the steric factor of the R_1_, R_2,_ R_3_ and R_4_ functional groups, and the presence of a methyl group with short alkyl chain in the structure of peptidomimetics 5 [50].

The results of our experiments are extremely important in understanding the new biological properties of the analyzed α-amidoamides 5 in the function of antibiotics and their toxic effect on gram-negative bacteria cells in the face of increasing bacterial resistance to various drugs.

## Figures and Tables

**Figure 1 materials-13-05169-f001:**
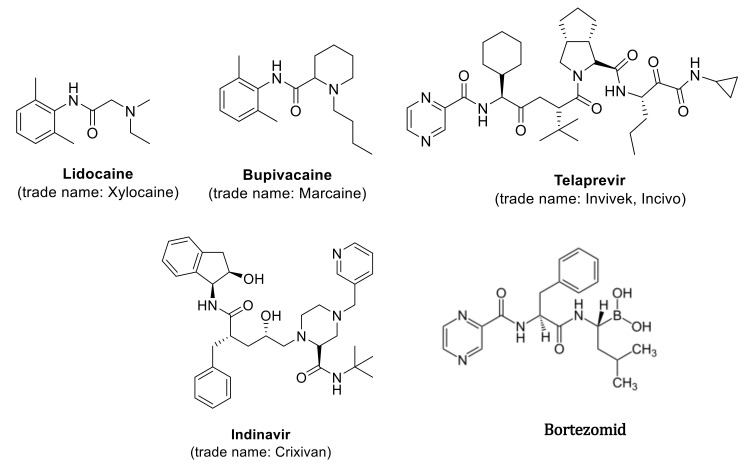
Examples of biologic active compound synthetized via Ugi reaction.

**Figure 2 materials-13-05169-f002:**
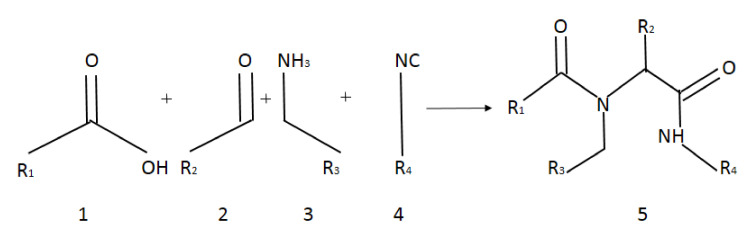
Scheme of Ugi reaction.

**Figure 3 materials-13-05169-f003:**
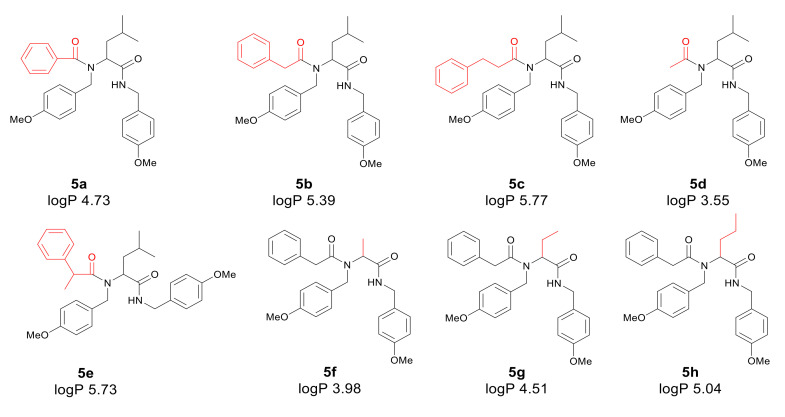
Products of the Ugi reaction. The compounds were isolated via column chromatography.

**Figure 4 materials-13-05169-f004:**
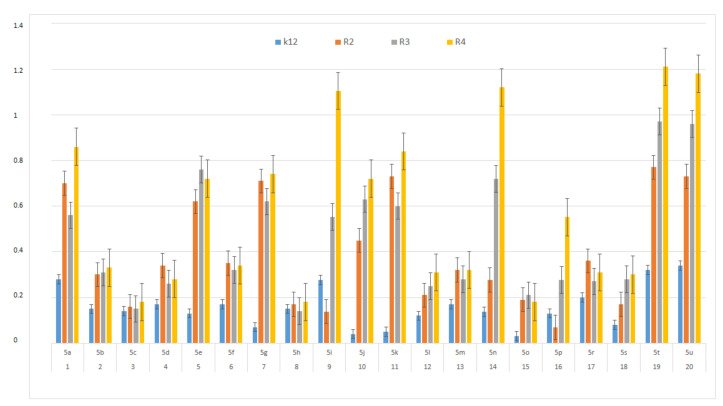
MIC of the α-amidoamids for the analysed strains of *Escherichia coli* K12 and R2, R2, and R4. The x-axis compounds **5a**–**5u** were used sequentially. The y-axis shows the MIC value in mg mL^−1^.

**Figure 5 materials-13-05169-f005:**
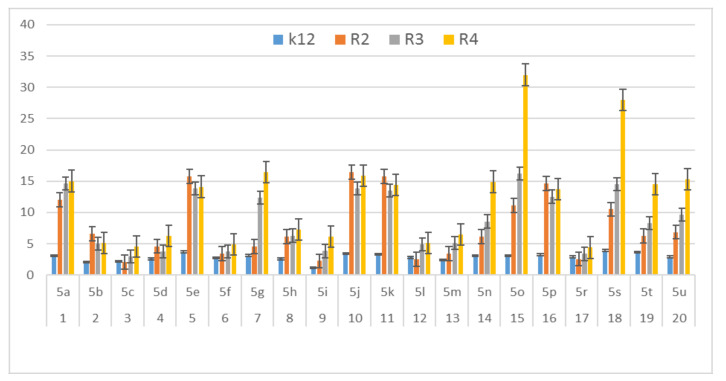
MBC of the α-amidoamids for the analysed strains of *Escherichia coli* K12 and R2, R2, and R4. The x-axis compounds **5a**–**5u** were used sequentially. The y-axis shows the MBC value in mg mL^−1^.

**Figure 6 materials-13-05169-f006:**
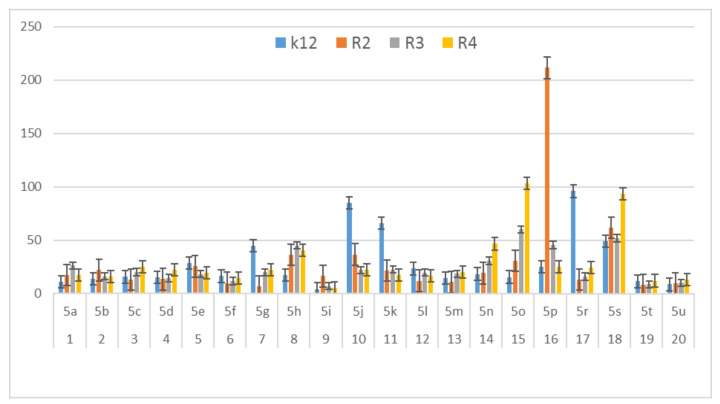
MBC/MIC of the α-amidoamids for the analysed strains of *Escherichia coli* K12 and R2, R2, and R4. The x-axis compounds **5a**–**5u** were used sequentially. The y-axis shows the MBC/MIC value in mg/mL^−1^.

**Figure 7 materials-13-05169-f007:**
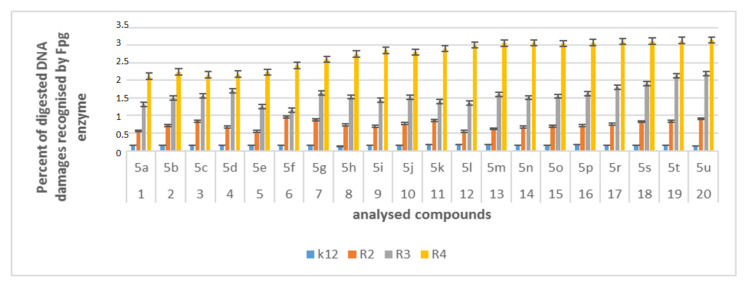
Percent of digested DNA damages recognised by Fpg enzyme- (y-axis) with control K12 and R2-R4 strains (x-axis); The compounds **5a**, **5b**, **5e**, **5g**, **5j**, **5k**, **5p,** and **5o** were statistically significant at <0.05 *.

**Figure 8 materials-13-05169-f008:**
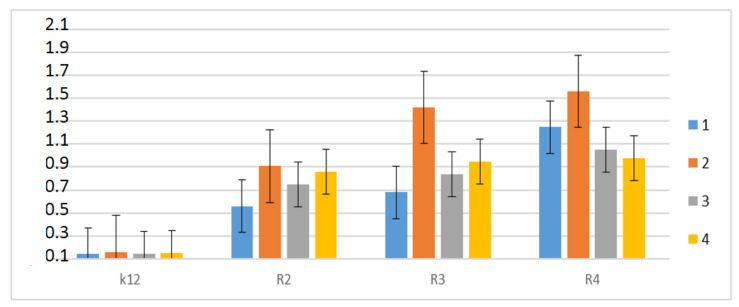
Examples of MIC with different strains K12, R2, R3, and R4 of the studied antibiotics with kanamycine (marked as 1-blue colour), bleomycine (marked as 2-orange colour), streptomycine (marked as 3-grey colour), and ciprofloxacine (marked as 4-yellow colour). The x-axis features antibiotics used sequentially. The y-axis features the MIC value in mM.

**Figure 9 materials-13-05169-f009:**
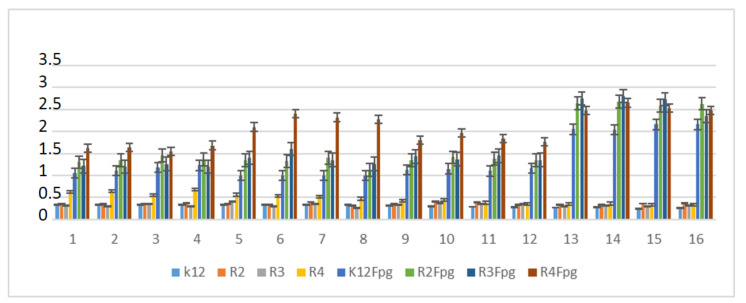
Percentage of plasmid DNA recognized by Fpg enzyme after ciprofloxacine (value 1–4), kanamycine (value 5–8), streptomycine (value 9–12), and bleomycine treatment (value 13–16). The compounds were statistically significant at <0.05 *.

**Table 1 materials-13-05169-t001:** Statistical analysis of 20 analyzed compounds by MIC, MBC, and MBC/MIC; <0.05 *, <0.01 **, <0.001 ***. MBC/MIC of the α-amidoamids ratio for the analyzed strains of *E. coli* K12 and R2–R4. α-amidoamids **5a**–**5u** were used sequentially.

No of Samples	5a	5b	5c	5d	5e	5f	5g	5h	5i	5j	5k	5l	5m	5n	5o	5p	5r	5s	5t	5u	Type of Test
K12	**				***		***			***	**			***		**			**	***	MIC
R2	**				***		***			***	**			***		**			**	***	MIC
R3	**				***		***			***	**			***		**			**	***	MIC
R4	**				***		***			***	**			***		**			**	***	MIC
K12	**				*		**			**	*			**		*			***	*	MBC
R2	**				*		**			**	*			**		*			***	*	MBC
R3	**				*		**			**	*			**		*			***	*	MBC
R4	**				*		**			**	*			**		*			***	*	MBC
K12	*				**		*			*	*			*		**			*	**	MBC/MIC
R2	*				**		*			*	*			*		**			*	**	MBC/MIC
R3	*				**		*			*	*			*		**			*	**	MBC/MIC
R4	*				**		*			*	*			*		**			*	**	MBC/MIC

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
