# Peer review of "α-Amidoamids as New Replacements of Antibiotics—Research on the Chosen K12, R2–R4 E. coli Strains"

_materials, 2020, doi:10.3390/ma13225169_

Round 1

Reviewer 1 Report

The manuscript is good written and interesting to read. However, I had few suggestion that will help to improve the over all manuscript.

  1. Introduction is too long. it distract the reader from the main theme of the manuscript. please reduce the lentgh to one and half page.
  2. There should be a graphical abstract which will increase the interes of the readers and will improve the understanding.
  3. please include a figure to show a biological activity of the amino acid on moecular level.
  4. conclusion is lengthy and is not properly written. please short it to the point.

Author Response

Thank you very much for all the extremely valuable suggestions that have contributed to the improvement of the scientific value of our work.

  • Introduction is too long. it distract the reader from the main theme of the manuscript. please reduce the lentgh to one and half page.
  •  
  • The introduction has been thoroughly redrafted and shortened to a page and a half

-There should be a graphical abstract which will increase the interes of the readers and will improve the understanding.

The graphic abstract was attached to the publication when the article was submitted. His version is quoted below.

  • please include a figure to show a biological activity of the amino acid on molecular level.

a biological activity of the amino acid on molecular level is presented as Ugi reaction on Figure 2 in manuscript, (in reference to a graphic abstract).

Figure 2. Scheme of Ugi reaction.

-conclusion is lengthy and is not properly written. please short it to the point.

Conclusions have been redrafted, changed, and presented in points in manuscript.

Reviewer 2 Report

Although an interesting concept, a lot of revisions need to be made to the manuscript including:

  1. The language and grammar need significant work. The spelling also needs to be checked, sometimes the compounds are referred to “aminoamids” and sometimes as “aminoamides”
  2. Parts of the introduction seem to repeat itself – starting with line 68. And the references seem to be inaccurate and sometimes cited doubly for very different topics.
  3. An introduction to the aminoamides would be appreciated as they are not really mentioned, but the intro focuses on aminoglycosides and beta-lactams.
  4. Even if the synthesis is unchanged from that in the literature, I recommend writing the general synthesis out.
  5. For the synthesis, did you differ R1? It is unclear but the figure seems to imply that. If the analogues have different scaffolds (as it seems 5a-5i and 5j-5u do), I would recommend separating them somehow i.e. 5a-5i and 6a-6j
  6. MICs and MBCs are concentrations, I really do not understand how you are reporting some values in mM. Also in the text you need to indicate what bacteria the MIC/MBC are in.
  7. Lines 437-494 should be in the discussion.
  8. I do not understand what you mean by different isocyanides, please define.
  9. How can you claim high toxicity to Gram-positive bacteria when you only looked at E.coli (line 492)? Remove if you did not test against any Gram-positives.
  10. The conclusion should more clearly define what substituents are best. I do not see any SAR.
  11. What group on the molecule do you think is crosslinking with DNA? I don’t see any reactive groups. You say aldehydes and isocyanides but those are not present in the actual molecule, only the reaction. I really hope you are not dosing the reaction.

Author Response

Thank you very much for all the extremely valuable suggestions that have contributed to the improvement of the scientific value of our work.

  1. The language and grammar need significant work. The spelling also needs to be checked, sometimes the compounds are referred to “aminoamids” and sometimes as “aminoamides”- has been revised and corrected.

- Language and grammar has been improved. The compound names "aminoamides" have been harmonized and corrected throughout the manuscript.

  1. Parts of the introduction seem to repeat itself – starting with line 68. And the references seem to be inaccurate and sometimes cited doubly for very different topics.

The introduction has been thoroughly redrafted and shortened to a page and a half (excluding the drawing in the introduction). Literature references have been checked and carefully assigned to the topics described

  1. An introduction to the aminoamides would be appreciated as they are not really mentioned, but the intro focuses on aminoglycosides and beta-lactams.

The short description on aminoamides is included without any focus on aminoglycosides and beta-lactams.

  1. Synthesis is unchanged from that in the literature, I recommend writing the general synthesis out.

The general synthesis contained in the manuscript.

  1. For the synthesis, did you differ R1? It is unclear but the figure seems to imply that. If the analogues have different scaffolds (as it seems 5a-5i and 5j-5u do), I would recommend separating them somehow i.e. 5a-5i and 6a-6j

α-amidoamids 5a-5u via the Ugi reaction between various carboxylic acids, aldehydes, amines and isocyanides in the presence of vesicles in aqueous medium were performed.

We ordered the substituents in the compounds from 5a to 5u to make their reactive groups more visible.

  1. MICs and MBCs are concentrations, I really do not understand how you are reporting some values in mM. Also in the text you need to indicate what bacteria the MIC/MBC are in.

Traditionally, the MIC and MBC values as defined in µg / ml or mg / l were in the manuscript in mM to show the potency of the toxic effect of the analyzed compounds on gram negative bacteria (intended by us) - however, upon the reviewer's suggestion, they were changed to generally accepted units. the relevant MICs and MBCs have been revised throughout the all manuscript.

  1. Lines 437-494 should be in the discussion.

-Lines 437-494 were moved to discussion.

  1. I do not understand what you mean by different isocyanides, please define.

Isocyanides (also called isonitriles) contain a nitrogen atom bonded to a carbon atom and an R group, with a resonance structure containing a triple bond, generating a carbanion and a positive nitrogen ion. Due to their unique reactivity, isocyanides are very popular components of a large number of organic reactions, particularly the Passerini and Ugi reactions. Isocyanide-based multicomponent reactions can be used to synthesize a diverse array of important compounds due to their functional group tolerance, as well as high levels of chemo-, regio-, and stereoselectivities. Isocyanides are valuable tools for the preparation of structurally diverse chemical libraries. Isocyanides comprise a diverse range of metabolites produced by terrestrial microorganisms, marine organisms, and plants. In our work, when defining various isocyanides, we think about the type of substituent in the alpha-aminoamide structure e.g. tetr-butyl isocyanide or 2,5-dimethoxybenzyl isocyanide.

  1. How can you claim high toxicity to Gram-positive bacteria when you only looked at E.coli (line 492)? Remove if you did not test against any Gram-positives.

Has been revised and corrected

  1. The conclusion should more clearly define what substituents are best. I do not see any SAR.

Conclusions have been redrafted, changed, and presented in points with the description of which substituent is the best.

  1. What group on the molecule do you think is crosslinking with DNA? I don’t see any reactive groups. You say aldehydes and isocyanides but those are not present in the actual molecule, only the reaction. I really hope you are not dosing the reaction.

The chemical compounds we use contain the group −N = C = O, in which the binding atom is the nitrogen atom. These can be salts of the type M − N = C = O (where M is a metal atom) or organic compounds of the type R − N = C = O (where R is any organic group, and the group −N = C = O is defined as isocyanate).They are very reactive compounds because they react with numerous nucleophiles in the cell, including nucleic acid base molecules in plasmids DNA.

All analyzed groups of compounds containing isocyanates and aldehydes can enter crosslinking reactions creating a steric hindrance for DNA, whereby the nucleic acid is modified with some bases, such as oxidized 8-oxoG, which is recognized by the Fpg enzyme. Fpg protein is also able to bind to modified DNA, and application of enzymes simultaneously enhances the potential formation of high molecular weight protein DNA-complex. This suggests that protein-DNA crosslinks formation via aldehyde groups of-adducts may be a ubiquitous mechanism; however, the extent of this phenomenon may be different for different proteins. Repair of analyzed three forms of DNA plasmids formed during incubation by four different antibiotics is realized at least by BER pathways. In our research we have found that system in E.coli is engaged in processing of replication forks similar to stalled by etheno DNA adducts after using repair glycosylase-Fpg. Similary, the formation of cyclic adducts by hydroxyalkenals is reversible; a N2-substituted linear guanine adduct with a free aldehyde group can be formed, which can in turn react with an amino group of the base from the opposite DNA strand to produce an interstrand crosslink. Such crosslinks, observed in oligodeoxynucleotide duplexes containing acrolein-dG adducts (Davi et al, 1999). Similary effect with interstrand crosslinks were observed in DNA treated with several other hydroxyalkenals, like crotonaldehyde and HNE, as well as with malondialdehyde, which bridges two DNA strands via aldehyde groups. To compare, the model acrolein adduct PdG strongly blocks DNA synthesis and presents a stronger block than g-OH-PdG when tested by primer extension experiments in vitro. This observation suggests that interstrand crosslinking has occurred, an event that would take place only if g-OH-PdG existed in a ring-open form similar to that reported for M1G paired with dC in duplex DNA (Mao et al., 1999).Fpg protein is also able to bind to modified DNA, and application of enzymes simultaneously enhances the potential formation of high molecular weight protein DNA-complex.  This suggests that protein-DNA crosslinks formation via aldehyde groups of-adducts may be a ubiquitous mechanism; however, the extent of this phenomenon may be different for different proteins. Repair of analysed three forms of DNA plasmids formed during incubation by four different antibiotics is realized at least by BER pathways. In our research we have found that system in E.coli is engaged in processing of replication forks similar to stalled by etheno DNA adducts after using repair glycosylase-Fpg.
We postulate that the interaction between DNA α-amidoamids and antibiotics involved covalent bonding between the aldehyde group and isocianide, which may be formed in single-stranded plasmid DNA, Each band represents the nucleotide position at which a break was induced by cleavage with DNA glycosylase/AP-lyase. Analysis of mapped the sites of enzymatic recognition suggest that all DNA residue are connected with modified oxidised bases which could correspond to 8-oxoguanine sites in the template. Probably base pair substituted with a heptyl side chain residue rearranged to lose the side chain, are efficiently excised by the Fpg glycosylase, and are new a substrate for them.

Reviewer 3 Report

In this manuscript, Paweł Kowalczyk et al. synthesized a series of new α-amidoamids by using Ugi reaction. The cytotoxic activity of these compounds was tested by MIC and MBC assay with E. coli R2, R3, R4, and K12 strains. Further, the role of these α-amidoamides on DNA modification was examined using agarose electrophoresis to compare the conformations change of the plasmid. Overall, this is a very interesting and well put-together study that validates the utility of massive screening for inhibitors of bacterial-specific DNA damage to expedite the discovery of antibiotics with novel modes of action. However, I have several questions regarding the experiments and the conclusions drawn:

  • The strategy of Ugi reaction is effective and practical, I do not doubt it. However, the statement on the results of MIC and MBC assay is somehow confusing. Usually, a good-working antibiotic would have a MIC value somewhere in the range of 1-10 ug/mL (or mg/L). Also, MIC values above 64 ug/mL are usually referred to as resistance to a particular drug. In lines 359,362,365,368 and 369, the author claimed that the best values of MIC the compounds range from 0.8 to 1.2mM, which could be converted to 400 to 600 ug/mL approximately. Most microbiologists would consider such values as resistance! In lines 372,375, the author claims the MIC value is 0.02mg/mL, however, this is a reasonable value. Any of these reported MIC values lie outside the reasonable MIC range, which means that these compounds cannot be used as antibiotics. Could this be due to an error in MIC determination or calculations? Regarding the data related to these results that can be seen in figure S1. The quality of the figures is very poor. The detailed label is missing as well. It is impossible to distinguish the color change of the wells. If the authors feel that I'm mistaken in my assessment of the field, a significant revision to supply the clear figure and the polt of the MIC &MBC calculation would be necessary to make this clear.
  • Regarding the DNA modification experiments, the quality of agarose gel is poor. I can not get useful information from it. Specific in S2, the author state in line 415," three traditional forms were observed", but the gel only shows one band. While the experiment is lacking the proper DNA ladder and the label of panels A and B. NO. 19 and 20 samples are missing in the down panel of the figure. In Fig. S4, which you have named as Figure 4, is lacking the proper DNA ladder again. Panel A seems an unexposed gel and it is hard to recognize the DNA bands. In Fig. S5, which you have named as Figure 5, the quality of DNA ladder is poor, a proper labeling of the molecular weight is needed to distinguish the bands. Figure 8 , unless I missed it, the differences among 1,2,3, 4 are not provided.
  • The author concluded that "α-amidoamids are able to modify all coli strains (R2 – R4) and their plasmid DNA, changing the spatial structure of LPS contained in their cell membrane."(line 623,624) . However, it is not clear that the data presented supports such an unequivocal statement. For example, do the authors have data demonstrating that α-amidoamids which they studied in this research could change the spatial structure of LPS? While, there are no directed functional assays on the contribution of these compounds to modify plasmid DNA.

Some minor points need to be addressed below.

  • Two paragraphs in lines 38-40 and 68-70, identical writing sentences should be avoided.
  • Line 65, Fig.3 is not corresponding to the statement here.
  • Line 76, reference should appear before the comma.
  • Line 77, Fig.3 is not corresponding to the statement here.
  • Lines 120-121, "They have an antibacterial effect by destroying the complex……" is not a precise statement. Reference 25 also not referred to it.
  • Line 130, "Escherichia coli" shold use Italic font.
  • Line 297, the duplication of "streptomycine"
  • Lines 301-303, the unit "ml" should be "mL".
  • Lines 365, "about 0.8mM for MIC and 15" missing the unit.
  • Lines 371, S2 in supplementary materials is not corresponding to the results here.
  • Line 403, "E. coli" should use Italic font.
  • Line 418, what is "S2 and 6" mean?
  • There are many "k12" (lowercase) in the manuscript, fig 3,4,5,6,7,8. Line 616. Stick to  "K12"(capital) and check to spell carefully,

Author Response

Thank you very much for all the extremely valuable suggestions that have contributed to the improvement of the scientific value of our work.

The strategy of Ugi reaction is effective and practical, I do not doubt it. However, the statement on the results of MIC and MBC assay is somehow confusing. Usually, a good-working antibiotic would have a MIC value somewhere in the range of 1-10 ug/mL (or mg/L). Also, MIC values above 64 ug/mL are usually referred to as resistance to a particular drug. In lines 359,362,365,368 and 369, the author claimed that the best values of MIC the compounds range from 0.8 to 1.2mM, which could be converted to 400 to 600 ug/mL approximately. Most microbiologists would consider such values as resistance! In lines 372,375, the author claims the MIC value is 0.02mg/mL, however, this is a reasonable value. Any of these reported MIC values lie outside the reasonable MIC range, which means that these compounds cannot be used as antibiotics. Could this be due to an error in MIC determination or calculations? Regarding the data related to these results that can be seen in figure S1.

The concentrations of the tested compounds used by our team, expressed in terms of MIC and MBC values, are high due to the fact that we wanted to check the effect at which the maximum dose of these compounds would be most effective. Currently, there is a view among microbiologists that the concentrations of antibiotics currently used, expressed by measuring the MIC and MBC in their action on bacteria, are "de facto" very low concentrations and may cause bacterial cell resistance. The concentrations of compounds used by us were many times higher and it was an effect intended to see if at the maximum concentrations - bacteria would respond to their cytotoxic activity. It is known that peptidomimetics, which include coumarin derivatives and the alpha-ammonamides currently studied by us, are, in a way, structural analogs of antibiotics. Their similar structure determines the toxicity to bacterial cells, while the biological effect is currently being investigated by us. Hence the search for new compounds and the analysis of the mechanisms of their penetration through the bacterial cell membrane (related to the length of the LPS). The mechanism of penetration of the "drug" into the cell seems to be crucial in order to induce oxidative stress for the bacterial cell and disintegrate its genetic material - hence the analysis of plasmid DNA and checking its substrate activity for the Fpg protein. Currently, the limit of increasing the concentration of drugs used for microorganisms is shifting towards a significant increase in their concentration in order to achieve the blown effect in a shorter time, as after a long-term action of antibiotics (treatments -7 days or longer). The MIC values we observed at the level of 0.02 mg as reasonable values result from a series of consecutive dilutions of the analyzed compounds, the initial concentrations of which have been standardized and show the trend of appropriate research. All MIC and MBC values have been recalculated and checked as suggested by the reviewer.

The quality of the figures is very poor. The detailed label is missing as well. It is impossible to distinguish the color change of the wells. If the authors feel that I'm mistaken in my assessment of the field, a significant revision to supply the clear figure and the polt of the MIC &MBC calculation would be necessary to make this clear.

In our manuscript, we included a better highlighted photo of the same analysed wells as for the MIC value. A detailed description is provided under the drawing.

Regarding the DNA modification experiments, the quality of agarose gel is poor. I can not get useful information from it. Specific in S2, the author state in line 415," three traditional forms were observed", but the gel only shows one band.

We improved the quality of the S2 gels by presenting as “positive” so that all forms of plasmid DNA- are better visible (arrows with form captions in the figure). The analysed plasmid forms isolated from the studied E.coli strains migrate in a characteristic way observed by us in previous experiments with coumarin derivatives [72]. The "oc" form migrates close to the edge of the sump while the linear form is very close to the "ccc" form. Hence, it is necessary to carry out longer electrophoresis to visualize all three analyzed forms - on the basis of which we wrote this statement in line 415.

  • While the experiment is lacking the proper DNA ladder and the label of panels A and B. NO. 19 and 20 samples are missing in the down panel of the figure

We did not give a ladder because we were interested in the effect of a given compound, not the size of its product. In addition, we only had a 20-well agarose gel comb, and we wanted the Fpg digestion reaction conditions for all modified trials to be performed under identical electrophoresis conditions, and to be quantified and compared with each other. We conducted several such experiments for individual strains. This one was shown as an example of enzyme digestion. Labels with the name of the panels have been added (thank you very much for this valuable suggestion). As for the lower panel of the drawing in which there are "samples", we explain that the samples were applied after digestion with the Fpg protein (the number of applied samples was qualitatively and quantitatively uniform in all wells - to maintain identical reaction conditions with regard to the concentration of DNA and the amount of FPG protein). However, the likely large number of modifications by compounds designated 19 and 20 resulted in the accumulation of numerous oxidative and alkylation base damage in the analysed DNA to the point where everything was digested by the FPG enzyme - visible as smudged spots. Which may be evidence of a large number of substituted and unsubstituted purines and the presence of modifications 7,8-dihydro-8-oxoguanine (8-oxoguanine), 8-oxoadenine, 4,6-diamino-5-formamidopyrimidines (Fapy-adenine), 2,6 -diamino- hydroxy-5-formamidopyrimidine (Fapy-guanine) and their derivatives - this was described in the discussion, lines 518-524. We observed a similar effect for both analysed compounds for the analysed MIC and MBC values.

In Fig. S4, which you have named as Figure 4, is lacking the proper DNA ladder again. Panel A seems an unexposed gel and it is hard to recognize the DNA bands.

As in Figure S2, also in Figure S4, for similar reasons, we did not give a ladder because we were interested in the effect of a given compound, not the size of its product. For better readability of the drawing, the gel was also shown in a positive form so that all forms of plasmid DNA were better visible (arrows with form captions in the figure).

In Fig. S5, which you have named as Figure 5, the quality of DNA ladder is poor, a proper labeling of the molecular weight is needed to distinguish the bands. Figure 8 , unless I missed it, the differences among 1,2,3, 4 are not provided.

We used a DNA ladder of 1500bp (the size of the 5 bold bands on the DNA ladder read from bottom to top are 200, 500, 1000, 1200 and 1500bp, respectively). Whereas the differences in Figure 8 are described in the results in lines 467-474.

The author concluded that "α-amidoamids are able to modify all coli strains (R2 – R4) and their plasmid DNA, changing the spatial structure of LPS contained in their cell membrane."(line 623,624) . However, it is not clear that the data presented supports such an unequivocal statement. For example, do the authors have data demonstrating that α-amidoamids which they studied in this research could change the spatial structure of LPS? While, there are no directed functional assays on the contribution of these compounds to modify plasmid DNA.

The Ugi reaction makes it possible to synthesize a huge amount of new chemical compounds using various ketones, aldehydes, acids and isonitriles. The obtained substances may turn out to be useful in the development of new pharmaceutically active substances. The only drawback is the lack of product variety, however combining this reaction with others increases the variety of products obtained. An example of combining the Ugi reaction with another reaction is the isoquinoline synthesis, in which the Heck reaction was used (Xiang etl a. 2004, Gedey et.al. 2002, Zhang et. al. 1999, Short et. al. 1997). Therefore, we are working on the development of such functional tests. Also based on our earlier experiments with coumarin derivatives and ionic liquids – citation [31,72,76] in manuscript. Where the mechanism of interaction with LPS of this type of compounds is described in detail.

[31] Kowalczyk, P.; Borkowski, A.; Czerwonka, G.; Cłapa, T.; Cieśla, J.; Misiewicz, A.; Borowiec, M.; Szala, M. The microbial toxicity of quaternary ammonium ionic liquids is dependent on the type of lipopolysaccharide. J. Mol. Liq. 2018, 266, 540–547, doi:10.1016/j.molliq.2018.06.102.

[72] Kowalczyk, P.; Madej, A.; Paprocki, D.; Szymczak, M.; Ostaszewski, R. Coumarin Derivatives as New Toxic Compounds to Selected K12, R1–R4 E. coli Strains. Materials 2020, 13, 2499. doi: 10.3390/ma13112499.

[76].Borkowski, A.; Kowalczyk, P.; Czerwonka, G.; Cieśla, J.; Cłapa, T.; Misiewicz, A.; Szala, M.; Drabik, M. Interaction of quaternary ammonium ionic liquids with bacterial membranes – Studies with Escherichia coli R1–R4-type lipopolysaccharides. Journal of Molecular Liquids 246 (2017) 282–289

Xiang, Z.; Luo, T.; Cui, J.; Shi, X.; Fathi, R.; Chen, J.; Yang, Z.. Novel Pd-II-mediated cascade carboxylative annulation to construct benzo[b]furan-3-carboxylic acids. „Organic Letters”. 6 (18), s. 3155–3158, 2004. DOI: 10.1021/ol048791n. PMID: 15330611.

Gedey, S.; Van der Eycken, J.; Fülöp, F.. Liquid-Phase Combinatorial Synthesis of Alicyclic β-Lactams via Ugi Four-Component Reaction. „Organic Letters”. 4 (11), s. 1967-1969, 2002. DOI: 10.1021/ol025986r.

Zhang, J.; Jacobson, A.; Rusche, J. R.; Herlihy, W.. Unique Structures Generated by Ugi 3CC Reactions Using Bifunctional Starting Materials Containing Aldehyde and Carboxylic Acid. „Journal of Organic Chemistry”. 64 (3), s. 1074-1076, 1999. DOI: 10.1021/jo982192a. PMID: 11674195.

Short K. M., Mjalli A. M. M.. A solid-phase combinatorial method for the synthesis of novel 5- and 6-membered ring lactams. „Tetrahedron Letters”. 38 (18), s. 359–362, 1997. DOI: 10.1021/ol048791n. PMID: 15330611.

-All minor point have been corrected and supplemented in the manuscript acc. reviewer's suggestions. All corrections are marked in azure.

  • Two paragraphs in lines 38-40 and 68-70, identical writing sentences should be avoided.
  • Line 65, Fig.3 is not corresponding to the statement here.
  • Line 76, reference should appear before the comma.
  • Line 77, Fig.3 is not corresponding to the statement here.
  • Lines 120-121, "They have an antibacterial effect by destroying the complex……" is not a precise statement. Reference 25 also not referred to it.
  • Line 130, "Escherichia coli" shold use Italic font.
  • Line 297, the duplication of "streptomycine"
  • Lines 301-303, the unit "ml" should be "mL".
  • Lines 365, "about 0.8mM for MIC and 15" missing the unit.
  • Lines 371, S2 in supplementary materials is not corresponding to the results here.
  • Line 403, "E. coli" should use Italic font.
  • Line 418, what is "S2 and 6" mean?
  • There are many "k12" (lowercase) in the manuscript, fig 3,4,5,6,7,8. Line 616. Stick to  "K12"(capital) and check to spell carefully,

Round 2

Reviewer 2 Report

Thank you for the edits you made, which I think improved the paper. The introduction especially is much better. There are still a few minor edits:

  1. I’m not sure what happened to the title but I think amidoamides got removed from it?
  2. The language and grammar are better but there are still a few mistakes; I recommend checking it again.
  3. The references are in different formats.
  4. Per previous point 11, I still do not see the group N=C=O in any of the compounds. I see all of them have two amides (N-C=0) but amides are not reactive. Again, I don’t understand what group is reacting. Can you please point out on one of the structures what group is crosslinking?

Author Response

Thank you for the edits you made, which I think improved the paper. The introduction especially is much better. There are still a few minor edits:

  1. I’m not sure what happened to the title but I think amidoamides got removed from it?

The title has been completed (sorry for this inconvenience)

  1. The language and grammar are better but there are still a few mistakes; I recommend checking it again.

the language and grammar were revised and revised throughout the manuscript also in terms of style and form

  1. The references are in different formats.

The references have been corrected and are in the same format.

  1. Per previous point 11, I still do not see the group N=C=O in any of the compounds. I see all of them have two amides (N-C=0) but amides are not reactive. Again, I don’t understand what group is reacting. Can you please point out on one of the structures what group is crosslinking?

We are very thankful for your comments. In all the cases the activity of tested compounds 5 was shown by the presence of different type of functional group R1, R2, R3 and R4. None of compound 5 contains isocyanide group while all these compounds were prepared from respective isocyanides R4-NC (Scheme 2). The structure of substrate isocyanide R4-CN determines the structure of α-amidoamines 5. So one can discuss this structure based on substrate used for reaction. That was not very fortunate. Therefore, in revised version of manuscript, the influence of the functional groups R1, R2, R3, R4, of target componds 5 on its biological activity is discussed.

Reviewer 3 Report

The manuscript has been improved significantly. The author clarified most of the questions properly. However, Some minor points still need to be addressed below.

  • Many thanks to the author supply reference 43. (Paweł Kowalczyk et al. 2018. The microbial toxicity of quaternary ammonium ionic liquids is dependent on the type of lipopolysaccharide. Journal of Molecular Liquids). The determination of MIC and MBC is clear to me now.

However, the figure quality of this manuscript still can be improved. In ref 43, which is published by the same author (Paweł Kowalczyk et al. 2018. Journal of Molecular Liquids). In this paper, a similar experiment was performed in figure 1 showing below, this is a good figure that presents the color change, concentration, and label of the microplate very clear. I think the author should not lower your standards, and maintain the equal quality of the figures as he did before.

                   Please find the figure in the attached pdf file     

  • Regarding the figures of agarose gel. The author claimed ” We did not give a ladder because we were interested in the effect of a given compound, not the size of its product. ” If so, why the author added a mark in figure S5 even poor quality of it. This answer lacks the basic principle of molecular biology experiments. The ladder is the most important positive control in the experiments. It not only indicates the MW of the product but also an index to verify the scientific reliability of the experiment. In this experiment, without the control of the ladder. It is impossible to quantify the bands in the gel. I even do not know if the band in the figure is the plasmid or contamination. In ref 43 (Paweł Kowalczyk et al. 2018. Journal of Molecular Liquids), figure 5 shows below, the author presents a proper gel figure and the gel quality is much better than the figures in this manuscript. There are no excuses, the author should add the ladder in all agarose figures to verify the reliability of the experiment.
  •  
  • Please find the figure in the attached pdf file    

  • Line 129,  "5kb-ladder (New England Biolabs) ", please supply the Cat. NO. I did not find this product from NEB.
  • Lines 372,375,378, 381,382 ,the unit "ml" should be "mL". please check all unit in the manuscript carfully.
  • The unit of MIC and MBC stick to ug mL-1 or mg mL-1 . For example, Lines 372,375,378, 381,382 is ug/ml , but in Line 385,395 is mg ml-1
  • Line 410 to 412, the figure 5 legend, “The y-axis the MBC/MIC value in µg/ml-1. ” This is an incorrect unit. As the author state in figure 4 and 5, “The y-axis the MBC and MIC value in mg ml-1”, how is possible the MBC/MIC value in µg/ml-1
  • Figure S5, the figure might from the print screen, please remove the wavy line and rotate button, and so on.

Author Response

The manuscript has been improved significantly. The author clarified most of the questions properly. However, Some minor points still need to be addressed below.

-Many thanks to the author supply reference 43. (Paweł Kowalczyk et al. 2018. The microbial toxicity of quaternary ammonium ionic liquids is dependent on the type of lipopolysaccharide. Journal of Molecular Liquids). The determination of MIC and MBC is clear to me now.

However, the figure quality of this manuscript still can be improved. In ref 43, which is published by the same author (Paweł Kowalczyk et al. 2018. Journal of Molecular Liquids). In this paper, a similar experiment was performed in figure 1 showing below, this is a good figure that presents the color change, concentration, and label of the microplate very clear. I think the author should not lower your standards, and maintain the equal quality of the figures as he did before.

                   Please find the figure in the attached pdf file     

Original photos of 48-well plates on which MIC tests were performed. It is known that the staining of individual wells on a plate is caused by the use of an appropriate compound that interacts with the cells of the analyzed bacterial strains. In the case of our previous publication, which used ionic liquids, the analyzed wells had a more "intense color" - this was due to the presence of quaternary ammonium groups reacting with bacterial cells and their subordinate theophylline groups with a different color range than the previously used ionic liquids, but it is known that that each compound reacts differently with the membrane of the bacterial cell and its genetic material.

In the future, in this type of experiment, I will try to better illuminate the plates to show the subtle differences in the intensity of the wells (so that there is no doubt about the observed effect we are talking about). Thank you very much to the Reviewer for pointing this to me.

R4 strain

R3 strain

R2 strain

K12 strain

  • Regarding the figures of agarose gel. The author claimed ” We did not give a ladder because we were interested in the effect of a given compound, not the size of its product. ” If so, why the author added a mark in figure S5 even poor quality of it. This answer lacks the basic principle of molecular biology experiments. The ladder is the most important positive control in the experiments. It not only indicates the MW of the product but also an index to verify the scientific reliability of the experiment. In this experiment, without the control of the ladder. It is impossible to quantify the bands in the gel. I even do not know if the band in the figure is the plasmid or contamination. In ref 43 (Paweł Kowalczyk et al. 2018. Journal of Molecular Liquids), figure 5 shows below, the author presents a proper gel figure and the gel quality is much better than the figures in this manuscript. There are no excuses, the author should add the ladder in all agarose figures to verify the reliability of the experiment.
  •  
  • Please find the figure in the attached pdf file   
  •  
  • The bands observed on the agarose gels are real and not impurity in any way. As I wrote earlier, plasmids of this type migrate in a very characteristic way. The oc form is very low and migrates just above the level of the agarose gel well, while the linear form and ccc migrate very close to each other. I agree that in all gels I should provide a ladder as a positive control to verify the experiment to determine the size of the analyzed bands - my mistake, the ladder is not presented - for which I apologize. However, as I wrote earlier, in specific experiments of this type, we were interested in how and whether the tested compounds react with plasmid DNA isolated from the analyzed strains (already used in earlier experiments with ionic liquids). At that time, work was carried out in the laboratory, where there were no problems with the availability of equipment. In the future I will try to use the ladder everywhere in this type of signage. And constantly improve the quality of the photos of agarose gels and MIC plates analyzed by me. The ladder that we gave in the experiments after modification with antibiotics and after preselection with selected alpha-amidamides served as a positive control, because in this particular experiment we were curious about the actual size of the analyzed product. The ladder used in this figure 5, which shows more pronounced bands, was taken from the second replicate with bleomycin on the gel and extrapolated to figure 5 for better verification of the bands and their size, and compared to a traditional commercial 1kb ladder which was taken from the NEB side. to show the size and mass of the analyzed fringes. In Figures S2 and S4, in the additional material, we used the extrapolation ladder from publications on ionic liquids and quoted at the Reviewer's request - in order to improve the quality of the analyzed results. In both cases the Quick -Load 1kb Extend DNA ladder was used

Thank you for the examples of how the presented results should look professional.

-All comments to line 129, 372,375,378, 381,382, 410 and 412 were corrected and marked in green in the manuscript.

-Figure S5 has been corrected and supplemented with a ladder for the correct verification of the experiment and a better reading of the bands on the gel.

  • Line 129,  "5kb-ladder (New England Biolabs) ", please supply the Cat. NO. I did not find this product from NEB.
  • Lines 372,375,378, 381,382 ,the unit "ml" should be "mL". please check all unit in the manuscript carfully.
  • The unit of MIC and MBC stick to ug mL-1 or mg mL-1 . For example, Lines 372,375,378, 381,382 is ug/ml , but in Line 385,395 is mg ml-1
  • Line 410 to 412, the figure 5 legend, “The y-axis the MBC/MIC value in µg/ml-1. ” This is an incorrect unit. As the author state in figure 4 and 5, “The y-axis the MBC and MIC value in mg ml-1”, how is possible the MBC/MIC value in µg/ml-1
  • Figure S5, the figure might from the print screen, please remove the wavy line and ..rotate button, and so on.
